# Zircon U-Pb Ages and Geochemistry of the Granite in the Xintianling Tungsten Deposit, SE China: Implications for Geodynamic Settings of the Regional Tungsten Mineralization

**Wu Yang, Min Zhang, Jun Yan and Xiaocui Chen \***

Faculty of Resources and Environmental Engineering, Guizhou Institute of Technology, Guiyang 550003, China; yangwu@git.edu.cn (W.Y.); zhangmin@git.edu.cn (M.Z.); yanjun@git.edu.cn (J.Y.)
\* Correspondence: chenxiaocui2007@163.com

**Abstract:** The Xintianling tungsten deposit is a super-large deposit in the Nanling tungsten–tin mineralization belt, which is genetically associated with the early-stage hornblende-biotite monzonitic granite of Qitianling pluton. The orebodies predominantly occur as veins and lenses within skarn rocks between Xintianling granite and limestone (Shidengzi group). In this work, whole-rock major and trace elements and zircon U–Pb ages of the Xintianling granite were studied in an attempt to investigate the geochronological framework, petrogenesis, tectonism, and metallogenesis with regard to the deposit. The petrographic and geochemical analyses indicated that the Xintianling granite consists of three intrusive units of medium- and coarse-grained biotite granite, fine-grained biotite granite, and granite porphyry, of which the biotite granite was strongly associated with mineralization. Biotite granite rocks are highly K-calc-alkaline and weakly peraluminous, with A/CNK ratios ranging from 0.99 to 1.05. Late-granite porphyry is aluminum-supersaturated with a high evolution degree, whose geochemical characteristics suggest that it is either an I- or S-type granite. LA-ICP-MS zircon U-Pb dating revealed that medium- and coarse-grained biotite granite ($162.3 \pm 1.2$ Ma, MSWD = 1.3), fine-grained biotite granite ($161.8 \pm 1.3$ Ma, MSWD = 1.8), and granite porphyry ($154.3 \pm 1.6$ Ma, MSWD = 2.4) formed in the late Jurassic. The emplacement of the Qitianling A-type granite and associated tungsten-tin polymetallic mineralization is a continuous evolution process, and they are products of the large-scale mineralization of the Nanling in the middle–late Jurassic (150–160 Ma). Under the tectonic setting of the Mesozoic lithospheric extension, asthenosphere upwelling along deep-fault, intensive mantle–crust interaction processes probably provide not only the high heat flow, but also partly mantle-derived material for large-scale W-Sn-polymetallic mineralization in this area.

**Keywords:** zircon U-Pb; granite geochemistry; genesis; Xintianling tungsten deposit

## 1. Introduction

The Nanling represents one of the most well-known, tungsten-rich areas in the world. The area is characterized by intensive magmatic activity. Previous geochronological studies have reported that, in this area, granite-related mineralization occurred from the Greenvillian to the late Yanshanlian period, peaking during 150~160 Ma [1–3]. The Xintianling deposit was discovered over 20 years ago and has 330,000 tons of W0$_3$, which is located in southern Hunan Province (e.g., the central part of the Nanling Range). Notably, it has always been acknowledged that Xintianling granite belongs to the Qitianling granite stock [4–8], but there are a limited number of studies regarding this granite. In particular, the petrogenic and geodynamic backgrounds of the Nanling tungsten–tin metallogenic belt in the Mesozoic era remain controversial. The debate centers on two main points. The first school of thought proposes that granite in the belt is closely related to S-type granite, whereas the other thought is that the exposed granite is primarily A-type granite.

To clarify the above ambiguity, we focused on tungsten-related granite in the Xintianling deposit in south China. We first presented the major and trace element compositions

of the granite to clarify its genesis. Then, zircon U-Pb dating was conducted to place geochronologic constraints on intrusions and related mineralization. Our new results show that the Xintianling tungsten deposit was formed in the late Jurassic under a back-arc extensional setting. The events were likely triggered by the break-off or detachment of the flatly subducted Paleo-Pacific slab (Figure 1).

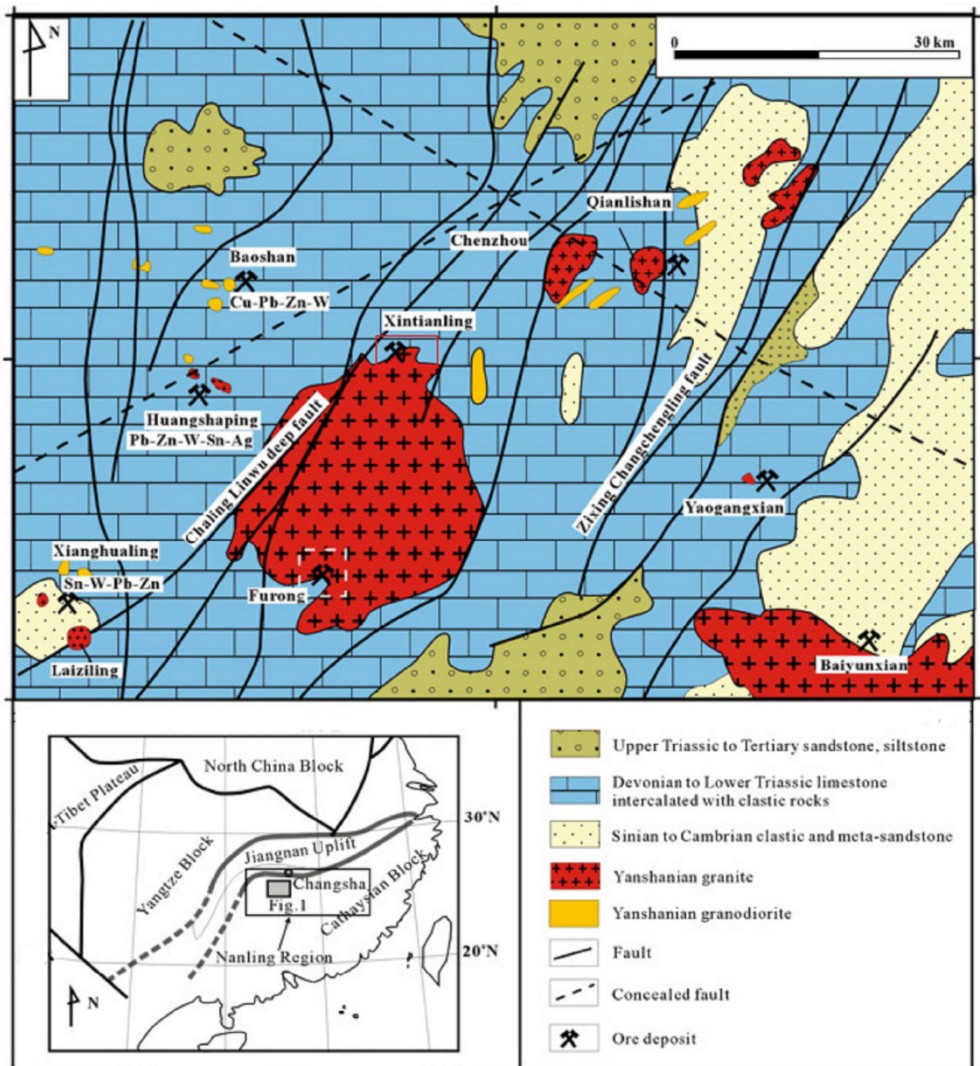

**Figure 1.** Simplified geological map of central Nanling region, South China, and distribution of mineral deposits in the region. (Peng et al. 2006) [5].

## 2. Regional Geological Characteristics

The Qianlishan–Qitianling research area has experienced many complicated tectonic movements at different stages and varying degrees, accompanied by magmatic activities. Its center is located where the Yangtze and the Cathaysian plates (the two major inland plates) converge and in the Chenzhou–Shaoyang strike-slip tectonomagmatic zone (Figure 2). The main area of the strike-slip tectonic zone lies within the Chenzhou–Shaoyang region, extending to the NW, to the Snowberg tectonic belt, but is outside Yaogangxian of Hunan Province in the southeast [9,10]. During that period, a significant ore-conducting structure developed in the area, controlling the distribution of ore fields with secondary structures controlling ore-body shapes and scales. The Yanling–Lanshan fault zone extends southwestward, from Yanling County to Langshan and through Chenzhou, with both ends stretching out through the province. It has a syndepositional foreland basin fault originating from the Sinian period (Early Paleozoic), a deep and large fault from the Caledonian movement

continuing until the Indo–Chinese–Yanshanian period, and a large-scale basement-cut fault with a long active period [11–14]. The Chenzhou–Shaoyang fault zone extends northwestward, from Yaogangxian through Chenzhou to Dayishan. It extends northwestward to the Snowberg arc structure belt at the center of Hunan and extends southeastward to Guangdong Indo–Chinese–Yanshanian in its important active period, causing the intrusion of intermediate-acid rocks from Guandimiao, Dayishan [14–18].

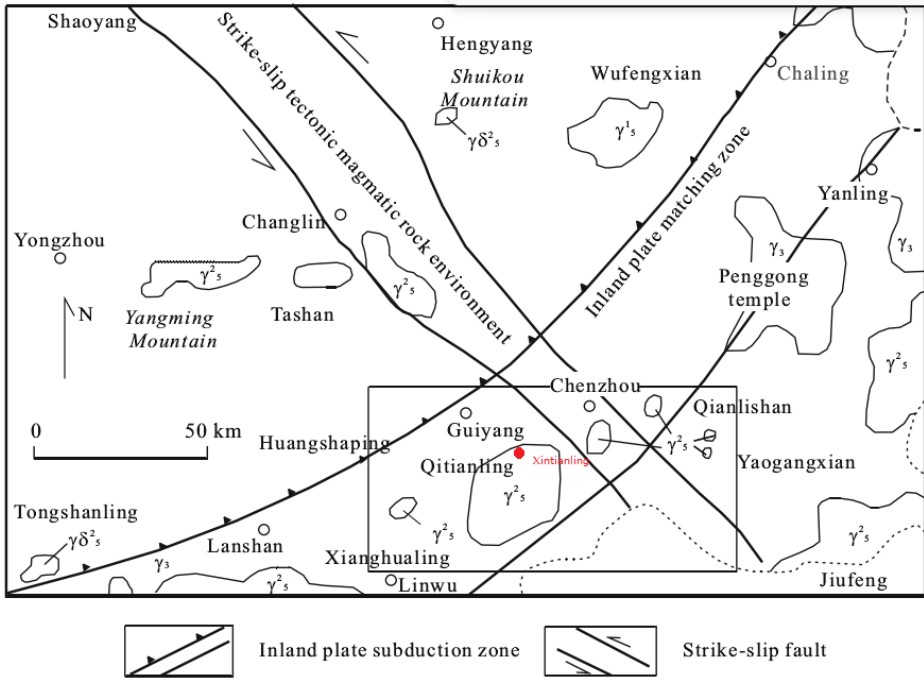

**Figure 2.** Tectonic setting of the Qianlishan–Qitianling area [7]. ($\gamma_3$.: Caledonian magmatic rock; $\gamma^1_5$: Indo-Chinese epoch magmatic rock; $\gamma^2_5$: Yanshanian magmatic rock; $\gamma\delta^2_5$: Yanshanian granodiorite).

## 3. Geologic Features of the Deposit

The Xintianling deposit is located north of the Qitianling pluton, which represents the central part of the Nanling Range. The strata are primarily carboniferous and covered locally by quaternary sediments. The carboniferous strata, from young to old, can be classified into the lower carboniferous YanGuan stage, the DaTang stage, and the middle–early Carboniferous TianHu stage. The exposed strata mainly consist of carbonates with silty and calcareous shale, and Permian carbonate rocks with ferromanganesian chert, siliceous and calcareous shale, which are unconformably overlain by Cretaceous red beds. Its mineralization is related to a north-south Duplex anticline. The main body of granite is micaceous granite, while the mineral grain size decreases from the center to the more peripheral zones of the pluton. It also can be divided into three lithologies: coarse-grained porphyritic biotite granite, medium- and coarse-grained porphyritic biotite granite, and fine- medium- and coarse-grained porphyritic biotite granite.

The main ore minerals present in the Xintianling deposit are scheelite and molybdenite, followed by small amounts of bismuthinite, galena, sphalerite, chalcopyrite, arsenopyrite, pyrrhotite, and pyrite. The main gangue minerals are garnet, diopside, actinolite, iron mica, chlorite, quartz, and small amounts of calcite, fluorite, epidote, etc. The main granular textures of the ores located here are automorphic, hypidiomorphic, and xenomorphic, as well as vein rocks. It should be noted that the main ore structures are disseminated structures. The main wall rock alterations that occur are skarnization silicification, greisenization, and marmarization. Notably, skarnization exhibits the most correlation with metallogenesis [19,20].

## 4. Sampling and Experimental Methods

The samples used in our study were collected from granite away from the mineralization in the Xintianling deposit. Petrographic observations (Figure 3) and chemical analysis (Table 1; see below) show that these samples suffered no hydrothermal alterations. We primarily selected two representative granites: medium- and coarse-grained porphyritic biotite granite and fine-grained biotite granite. Some granitic porphyries were tested in supplementary analyses (Figure 4).

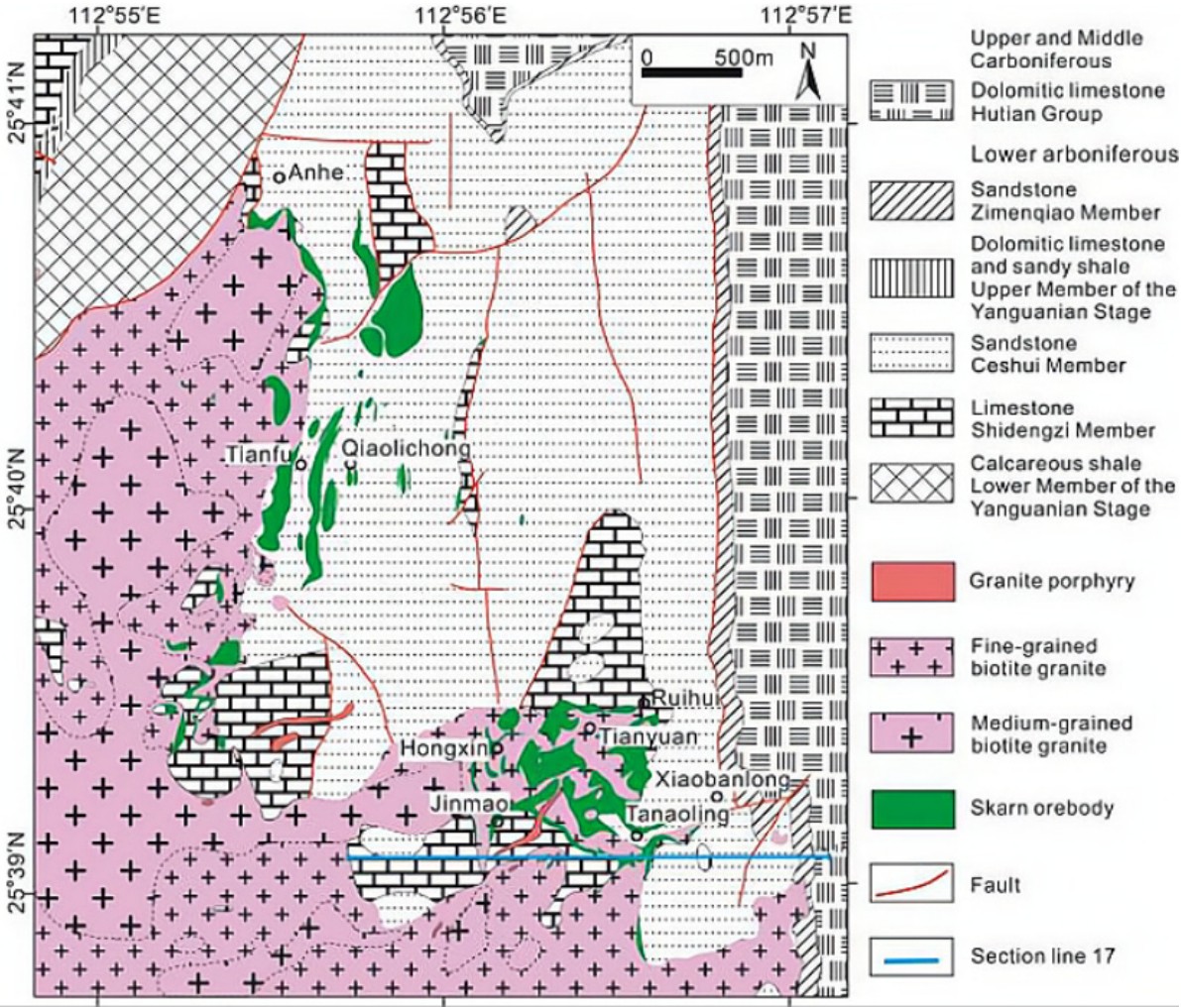

**Figure 3.** Sketch geological map of the Xintianling deposit (Zhang et al., 2015) [8].

**Table 1.** Main elements of granite in Xintianling ore field, Hunan (wt%).

| SAMPLE | | SiO$_2$ | Al$_2$O$_3$ | Fe$_2$O$_3$ | CaO | MgO | K$_2$O | Na$_2$O | TiO$_2$ | MnO | P$_2$O$_5$ | BaO | SrO | LOI |
|---|---|---|---|---|---|---|---|---|---|---|---|---|---|---|
| XTL-01 | | 70.1 | 14.25 | 3.19 | 2.29 | 0.8 | 4.66 | 2.95 | 0.45 | 0.06 | 0.18 | 0.11 | 0.03 | 0.72 |
| XTL-07 | Medium- | 66.1 | 15.4 | 4.54 | 2.66 | 1.18 | 4.34 | 3.34 | 0.65 | 0.09 | 0.27 | 0.1 | 0.04 | 1.12 |
| XTL398 | grained | 67.7 | 13.95 | 3.35 | 2.89 | 1.31 | 4.31 | 2.46 | 0.43 | 0.13 | 0.17 | 0.1 | 0.03 | 2.17 |
| XTL-250-3 | porphyritic | 75.7 | 12.55 | 1.97 | 0.93 | 0.17 | 5.15 | 2.82 | 0.11 | 0.05 | 0.01 | 0.02 | 0.01 | 0.98 |
| XTL-200-1 | biotite | 68.6 | 14.2 | 3.61 | 2.04 | 0.84 | 4.63 | 3.09 | 0.43 | 0.08 | 0.18 | 0.13 | 0.03 | 1.2 |
| JY-468-3 | granite | 72.4 | 12.75 | 2.28 | 1.82 | 0.53 | 5.54 | 2.03 | 0.23 | 0.06 | 0.09 | 0.05 | 0.02 | 2.18 |
| AH-365-1-2 | | 73.6 | 13.4 | 2.05 | 1.26 | 0.47 | 4.66 | 3.36 | 0.23 | 0.05 | 0.09 | 0.04 | 0.02 | 0.92 |
| XTL-03 | Fine-grained | 73.8 | 13 | 1.93 | 0.99 | 0.42 | 5.52 | 2.92 | 0.22 | 0.06 | 0.07 | 0.06 | 0.02 | 0.87 |
| JY-415-01 | biotite | 74.9 | 13.05 | 1.14 | 1.23 | 0.19 | 4.61 | 3.44 | 0.11 | 0.07 | 0.02 | 0.02 | 0.01 | 0.88 |
| AH-365-1-1 | granite | 74.1 | 13.4 | 1.18 | 0.95 | 0.21 | 5.21 | 3.67 | 0.11 | 0.03 | 0.05 | 0.02 | 0.01 | 0.73 |
| XT-160-3 | | 76.3 | 12.45 | 0.98 | 1.09 | 0.08 | 4.32 | 3.59 | 0.06 | 0.03 | 0.01 | 0.02 | 0.01 | 0.93 |

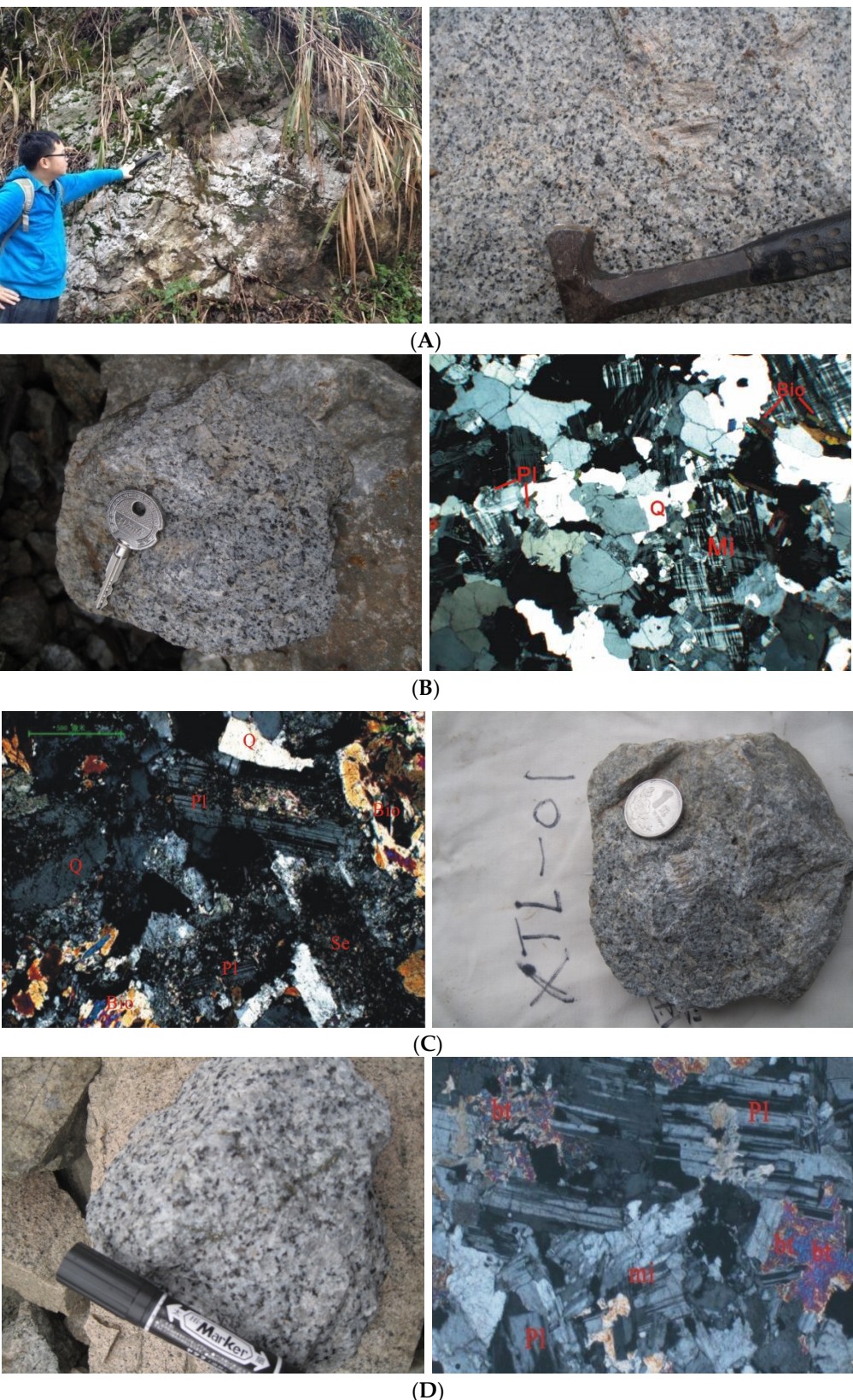

**Figure 4.** Field, macroscopic, and microscopic features of Xintianling granite. (**A**) Photos of granite outcrops in the wild; (**B**) Field hand specimen of granite porphyry and its microscopic characteristics (5 × 10 orthogonal polarized light); (**C**) Field hand specimen of granite porphyry and its microscopic characteristics (5 × 10 orthogonal polarized light); (**D**): Field hand specimens and microscopic features of pyritized biotite granite (5 × 10 orthogonal polarized light).

The major element abundances were determined using X-ray fluorescence, utilizing a Panalytical (formerly Philips Analytical Instruments Division) Model Axios (PW4400) spectrometer. The trace and rare-earth element abundances were analyzed via inductively coupled plasma mass spectrometry (ICP-MS) using a Finnigan MAT Model ELEMENT. The analyses were conducted in an ALS mineral laboratory in China. The detailed analytical procedures were previously described by Liu [4]. Two biotite granite samples (XTL-200-01-03 and XTL-340-7) and one granite porphyry sample (XTL-06) were selected for use in zircon dating. First, the fresh sample was crushed to a size of 200 meshes. Next, the zircon was selected under a binocular microscope. Target preparation and testing were subsequently performed based on this. The sample crushing, zircon selection, and target preparation were completed at the laboratory of Hebei Institute of Regional Geological and Mineral Survey, China. Analyses of zircon trace element contents and U-Pb isotope dating were simultaneously performed via LA-ICP-MS at the State Key Laboratory of Geological Processes and Mineral Resources, China University of Geosciences (Wuhan). The detailed instrument operating conditions and data-processing methods were identical to those used by Liu [4] and are briefly summarized below. We used the GeoLas 2005 laser ablation system, whereas the ICP-MS was conducted using Agilent 7500a. Helium and argon were used as the carrier and compensating gases, respectively, to adjust the sensitivity during the laser ablation process. The two gases were mixed via a T-joint before entering the ICP. A small amount of nitrogen was added to the plasma central gas flow (Ar + He) to enhance the instrument sensitivity, lower the detection limit, and increase the analytical precision. We calibrated the data with a standard sample during the test. Detailed instructions are given in the data analysis section.

## 5. Results

### 5.1. Geochemistry

After petrographic examination and the removal of altered surfaces, the samples for the whole-rock analysis were crushed in an agate mill to 200 mesh. X-ray fluorescence (XRF; Rigaku RIX 2100 spectrometer) using fused glass disks and ICP-MS (Agilent 7500a with a shield torch) was used to measure the major and trace-element compositions, respectively, of ALS Minerals (ALS Chemex) after the acid digestion of samples in Teflon bombs. The analytical precision was better than 5% for major elements and often better than 10% for trace elements [21]. The detailed analytical procedures for major-element analysis by XRF are described in [22], while those for trace element analysis by ICP-MS are described in [23].

The results of the analyses of major and trace elements are listed in Tables 1–5.

**Table 2.** Trace elements of granite in Xintianling ore field, Hunan (ppm).

| SAMPLE | | Rb | Cs | Ba | Hf | Th | U | Ta | Nb | W | Sr | Zr |
|---|---|---|---|---|---|---|---|---|---|---|---|---|
| XTL-01 | | 269 | 13.65 | 951 | 6.1 | 27.1 | 9.34 | 2.9 | 24.8 | 2 | 306 | 205 |
| XTL-07 | | 224 | 10.65 | 922 | 8 | 39.9 | 8.77 | 2.6 | 27.8 | 2 | 416 | 289 |
| XTL398 | Medium-grained | 282 | 31.7 | 865 | 5.9 | 24.3 | 6.72 | 1.6 | 21.1 | 14 | 282 | 212 |
| XTL-250-03 | porphyritic biotite | 523 | 13.65 | 84.8 | 6.5 | 94.7 | 29.1 | 4.7 | 32.7 | 10 | 74.9 | 158 |
| XTL-200-1 | granite | 215 | 7.49 | 1180 | 6.6 | 30.4 | 4.63 | 1.7 | 20.7 | 34 | 251 | 254 |
| JY-468-3 | | 291 | 18.45 | 394 | 4 | 33.9 | 16.8 | 1.5 | 22.6 | 30 | 141 | 140 |
| AH-365-1-2 | | 309 | 16.7 | 327 | 3.9 | 27.3 | 15.65 | 4.9 | 26.1 | 8 | 188 | 111 |
| XTL-03 | | 300 | 19.8 | 517 | 3.4 | 34.7 | 16.25 | 2.3 | 19.4 | 3 | 206 | 104 |
| JY-415-01 | Fine-grained biotite | 404 | 51.5 | 84.3 | 3.3 | 29.1 | 28.1 | 3.3 | 24.4 | 18 | 81.1 | 72 |
| AH-365-1-1 | granite | 427 | 14.7 | 163.5 | 3 | 19.65 | 23.5 | 11.3 | 45.9 | 9 | 106.5 | 53 |
| XT-160-3 | | 346 | 19.65 | 87 | 3.3 | 25.1 | 44.3 | 6.9 | 43.4 | 28 | 48.5 | 53 |

**Table 3.** Rare-earth element concentrations in granite in Xintianling ore field, Hunan (ppm).

| SAMPLE | | La | Ce | Pr | Nd | Sm | Eu | Gd | Tb | Dy | Ho | Er | Tm | Yb | Lu | Y |
|---|---|---|---|---|---|---|---|---|---|---|---|---|---|---|---|---|
| XTL-01 | | 57.9 | 113 | 12.15 | 41.5 | 7.76 | 1.4 | 5.98 | 0.87 | 4.49 | 0.83 | 2.32 | 0.36 | 2.3 | 0.33 | 24.8 |
| XTL-07 | | 95 | 181 | 18.95 | 61.9 | 11.4 | 1.79 | 8.35 | 1.17 | 5.88 | 1.13 | 3.1 | 0.47 | 2.75 | 0.47 | 33.2 |
| XTL398 | Medium- | 49.2 | 91.4 | 9.5 | 34.4 | 6.11 | 1.19 | 4.35 | 0.6 | 3.36 | 0.54 | 1.6 | 0.23 | 1.43 | 0.22 | 16.7 |
| XTL-250-03 | grained | 73.5 | 155 | 16.4 | 51.2 | 11.1 | 0.33 | 9.71 | 1.79 | 10.95 | 2.41 | 7.2 | 1.19 | 7.93 | 1.22 | 73.2 |
| XTL-200-1 | porphyritic | 73 | 127.5 | 13.15 | 45.1 | 7.82 | 1.54 | 6.52 | 0.85 | 5.21 | 0.96 | 2.95 | 0.39 | 2.57 | 0.37 | 27.9 |
| JY-468-3 | biotite granite | 29 | 50.5 | 5.03 | 16.7 | 2.85 | 0.59 | 2.47 | 0.37 | 2.25 | 0.43 | 1.44 | 0.22 | 1.45 | 0.26 | 16.3 |
| AH-365-1-2 | | 25.4 | 51.9 | 5.55 | 19.1 | 4.14 | 0.78 | 3.47 | 0.59 | 3.49 | 0.71 | 2.09 | 0.33 | 2.5 | 0.41 | 24.1 |
| XTL-03 | | 41.4 | 77.1 | 7.73 | 23.5 | 4.12 | 0.73 | 2.89 | 0.49 | 2.57 | 0.57 | 1.5 | 0.24 | 1.81 | 0.32 | 16.6 |
| JY-415-01 | Fine-grained | 12.3 | 26.2 | 3.03 | 12.3 | 3.04 | 0.49 | 2.89 | 0.44 | 2.69 | 0.58 | 2.1 | 0.27 | 2.27 | 0.41 | 19.1 |
| AH-365-1-1 | biotite granite | 13.9 | 29.6 | 3.43 | 11.9 | 3 | 0.43 | 2.92 | 0.53 | 3.25 | 0.67 | 2 | 0.35 | 2.57 | 0.45 | 21.9 |
| XT-160-3 | | 6.9 | 17.8 | 2.43 | 10.9 | 3.84 | 0.49 | 4.28 | 0.79 | 5.51 | 1.12 | 3.75 | 0.55 | 3.93 | 0.64 | 31.7 |

**Table 4.** Rare-earth element characteristics in granite in Xintianling ore field, Hunan.

| SAMPLE | | ΣREE | LREE | HREE | w(LREE)/w(HREE) | LaN/YbN | δEu | δCe |
|---|---|---|---|---|---|---|---|---|
| XTL-01 | | 251.19 | 233.71 | 17.48 | 13.37 | 18.06 | 0.60 | 0.99 |
| XTL-07 | | 393.36 | 370.04 | 23.32 | 15.87 | 24.78 | 0.54 | 0.99 |
| XTL398 | Medium-grained | 204.13 | 191.80 | 12.33 | 15.56 | 24.68 | 0.67 | 0.97 |
| XTL-250-03 | porphyritic biotite | 349.93 | 307.53 | 42.40 | 7.25 | 6.65 | 0.09 | 1.05 |
| XTL-200-1 | granite | 287.93 | 268.11 | 19.82 | 13.53 | 20.37 | 0.64 | 0.93 |
| JY-468-3 | | 113.56 | 104.67 | 8.89 | 11.77 | 14.35 | 0.66 | 0.94 |
| AH-365-1-2 | | 120.46 | 106.87 | 13.59 | 7.86 | 7.29 | 0.61 | 1.02 |
| XTL-03 | | 164.97 | 154.58 | 10.39 | 14.88 | 16.41 | 0.61 | 0.98 |
| JY-415-01 | Fine-grained | 69.01 | 57.36 | 11.65 | 4.92 | 3.89 | 0.50 | 1.02 |
| AH-365-1-1 | biotite granite | 75.00 | 62.26 | 12.74 | 4.89 | 3.88 | 0.44 | 1.02 |
| XT-160-3 | | 62.93 | 42.36 | 20.57 | 2.06 | 1.26 | 0.37 | 1.06 |

**Table 5.** CTPW standard minerals in granite in Xintianling ore field, Hunan, China.

| Parameters\Sample | XTL-01 | XTL-03 | XTL-07 | XTL398 | XTL-250-03 | JY-415-01 | XT-200-1 | XTL-160-3 | JY-468-3 | AH-365-1-1 | AH-365-1-2 | AH-256-2 |
|---|---|---|---|---|---|---|---|---|---|---|---|---|
| Quartz (Q) | 31.67 | 34.4 | 23.26 | 31.6 | 37.17 | 34.78 | 28.56 | 37.16 | 37.1 | 30.91 | 33.02 | 5.78 |
| Calcium feldspar (An) | 10.54 | 4.66 | 11.93 | 13.89 | 4.62 | 6.09 | 9.47 | 5.43 | 8.73 | 4.5 | 5.84 | 0 |
| Sodium feldspar (Ab) | 21.09 | 22.8 | 27.76 | 17.8 | 23.88 | 29.07 | 25.14 | 29.83 | 14.51 | 31.07 | 28.49 | 44.34 |
| Potash feldspar (Or) | 27.71 | 32.9 | 26.04 | 26.24 | 30.63 | 27.58 | 27.99 | 25.79 | 33.39 | 31.13 | 27.79 | 1.77 |
| Corundum (C) | 1.3 | 0.94 | 1.1 | 1.01 | 0.68 | 0.28 | 1.04 | 0.06 | 0.86 | 0.16 | 0.75 | 0 |
| Hypersthene (Hy) | 3.39 | 1.89 | 5.02 | 5.28 | 1.5 | 1.1 | 3.89 | 0.74 | 2.52 | 1.03 | 2.11 | 13.37 |
| Ilmenite (Il) | 0.86 | 0.42 | 1.25 | 0.84 | 0.21 | 0.21 | 0.84 | 0.12 | 0.45 | 0.21 | 0.44 | 0.45 |
| Magnetite (Mt) | 1.92 | 1.24 | 2.72 | 1.93 | 1.24 | 0.72 | 2.21 | 0.62 | 1.4 | 0.78 | 1.28 | 0 |
| Apatite (Ap) | 0.42 | 0.16 | 0.64 | 0.41 | 0.02 | 0.05 | 0.43 | 0.02 | 0.21 | 0.12 | 0.21 | 0.12 |
| Zircon (Zr) | 0.04 | 0.02 | 0.06 | 0.04 | 0.03 | 0.01 | 0.05 | 0.01 | 0.03 | 0.01 | 0.02 | 0.02 |
| Chromite (Cm) | 0 | 0.01 | 0.01 | 0 | 0 | 0 | 0 | 0 | 0.01 | 0 | 0 | 0.01 |
| Total | 100.03 | 100.01 | 100.04 | 100.03 | 100.02 | 100 | 100.06 | 100.01 | 100.02 | 100.01 | 100 | 100.03 |
| Differentiation index (DI) | 80.47 | 90.1 | 77.06 | 75.64 | 91.68 | 91.43 | 81.69 | 92.78 | 85 | 93.11 | 89.3 | 51.89 |
| Density g/cc | 2.7 | 2.66 | 2.73 | 2.71 | 2.66 | 2.64 | 2.7 | 2.64 | 2.67 | 2.64 | 2.66 | 2.83 |
| Liquid density | 2.41 | 2.37 | 2.45 | 2.43 | 2.37 | 2.36 | 2.42 | 2.36 | 2.39 | 2.36 | 2.38 | 2.67 |
| Dry viscosity | 8.72 | 10.38 | 7.26 | 8.35 | 11.31 | 11.16 | 8.45 | 11.79 | 10.25 | 10.59 | 10.28 | 5.32 |
| Wet viscosity | 6.62 | 7.31 | 5.88 | 6.47 | 7.72 | 7.7 | 6.5 | 7.9 | 7.3 | 7.44 | 7.33 | 4.7 |
| Liquidus temperature | 839 | 767 | 902 | 853 | 734 | 740 | 846 | 717 | 778 | 757 | 769 | 1004 |
| H₂O (%) | 3.5 | 4.26 | 2.75 | 3.33 | 4.57 | 4.51 | 3.37 | 4.78 | 4.16 | 4.32 | 4.18 | 1.73 |
| A/CNK | 1.013 | 1.033 | 1.025 | 0.999 | 1.054 | 1.013 | 1.029 | 0.991 | 1.008 | 1 | 1.042 | 0.245 |
| Consolidation index (SI) | 7.01 | 3.93 | 8.99 | 11.67 | 1.7 | 2.04 | 7.02 | 0.9 | 5.17 | 2.06 | 4.51 | 1.63 |
| Alkalinity rate (AR) | 2.7 | 4.04 | 2.48 | 2.34 | 3.89 | 3.58 | 2.81 | 3.81 | 3.16 | 4.25 | 3.42 | 1.02 |
| Composite index σ43 | 2.13 | 2.31 | 2.52 | 1.82 | 1.94 | 2.02 | 2.3 | 1.87 | 1.93 | 2.53 | 2.09 | 0.01 |
| Composite index σ25 | 1.29 | 1.47 | 1.44 | 1.09 | 1.26 | 1.31 | 1.38 | 1.23 | 1.22 | 1.61 | 1.33 | 0 |

### 5.1.1. Major Elements

The results of the analysis of major elements are shown in Table 1. The $SiO_2$ content of the Xintianling granite varies from 66.1 wt% to 76.3 wt%, with an average of 72.11 wt%. The average $SiO_2$ content in medium- and coarse-grained porphyritic biotite granite is 70.6 wt%, whereas that of the fine-grained biotite granite is 74.78 wt%. Medium- and coarse-grained porphyritic biotite granite exhibits higher (MgO + CaO) contents with TFe > 1.00 wt% than fine-grained biotite granite. The $K_2O$ and $Na_2O$ contents in all the samples range from 4.31 wt% to 5.54 wt% and 2.03 wt% to 3.67 wt%, respectively ($K_2O$ + $Na_2O$ > 7.00%). It was found that $K_2O/Na_2O$ is high (1.20–2.73). The $Al_2O_3$ contents are generally high in the samples, as they range from 12.45 wt% to 15.4 wt%, with an average of 13.49 wt%.

### 5.1.2. Trace Elements

The results of the analysis of trace elements are shown in Table 2. The mass fractions of large-ion lithophile elements (Rb, Cs, and Sr) were found to be stable. In most samples, the contents of Rb, Cs, and Sr varied from 215 to 346 ppm, 10.65 to 19.8 ppm, and 106.5 to 306 ppm, respectively. However, the large-ion lithophile element Ba exhibited a remarkable variability, fluctuating from 84.3 to 1180 ppm. The ratio of Sr/ Ba was somewhat low, at 0.32–0.96. Moreover, non-active elements (Nb, Ta, Zr, and Hf) exhibit high contents, which vary within the ranges of 19.4–45.9 ppm, 1.5–11.3 ppm, 53–289 ppm, and 3–8 ppm, respectively. Notably, the Nb/Ta values range from 4.06 to 13.18. Xintianling granite is shown to be enriched with Rb, Sr, U, and other large-ion lithophile elements, while the levels of Nb, Th, and other high-field-strength elements are somewhat low.

### 5.1.3. REE

The results of the analysis of trace elements are shown in Table 3, while the statistical characteristics of the REEs are summarized in Table 4. As seen in Table 4, light REEs enriched granite at relatively high levels, as the LREE values range from 42.36 ppm to 370 ppm. It was found that the HREE values vary from 8.89 ppm to 42.4 ppm. At the same time, the LREE/HREE values are in the range of 2.06–15.87 and the ΣREE values vary from 69.93 ppm to 393 ppm, signifying low values. The δEu value is equal to 0.09−t 0.67, the δCe value is 0.93–1.06, and the $(La/Yb)_N$ value is higher in most samples (6.65–24.78). The rock mass CIPW calculation results are summarized in Table 5. As can be seen, the mass fractions of An, Ab, Or, Hy, Mt, and Zr vary within the ranges of 4.5–13.89, 14.51–29.07, 26.04–33.39, 0.74–5.28, and 0.62–2.72, respectively. The main rock-forming minerals are quartz, calcium feldspar, sodium feldspar, and orthoclase. Accessory minerals include corundum, hypersthene, magnetite, phosphorus pyroxene, zircon, chromite, etc. The mass fraction of the standard mineral corundum is generally low, i.e., it ranges between 0.16 and 1.3. Furthermore, it was found that the alkalinity rate is within the 2.34–4.25 range. Moreover, the solidification index was found to be between 0.9 and 11.67, with an average of 5. The aluminum saturation index was estimated to be within the 0.991–1.054 range, with a mean of 1.019 (<1.10). This finding demonstrates that rock belongs to the high K-Calc-alkaline (HKCA) series [24,25].

### 5.2. Zircon U-Pb Geochronology

Zircons were isolated from medium-grained porphyritic biotite granite and fine-grained biotite granite using combined magnetic and heavy liquid separation techniques at the Geological Laboratory of the Regional Geological Survey, Langfang City, Hebei Province, China. The zircon grains were examined under transmitted and reflected light using an optical microscope. Distinct domains within the zircons were selected for analysis based on their CL images. An Agilent 7500a ICP-MS equipped with a 193-nm laser was used. In the experiment, high-purity He was used as the carrier gas of the ablated substance. The laser operating frequency was 10 Hz, the laser spot diameter at the test point was 36 μm, and the effective acquisition time of the mass spectrometer was 45 s. U–Pb isotope fractionation uses the international standard zircon 91,500 as the external correction and

TEM (416 ± 5 Ma) and QH (160 ± 1 Ma) as monitoring standards. Samples housed at the State Key Laboratory of Geological Processes and Mineral Resources, China University of Geosciences (Wuhan), were used to measure the U–Pb ages of zircons. The ICP-MS DataCal (Ver. 6.7) [24] and Isoplot (Ver. 3.0) [25] programs were used for data reduction [26]. The correction for common Pb was performed following [27]. The dating results are presented in Tables 6–8.

**Table 6.** LA-ICPMS U–Pb zircon dating data of the XTL200 from the Xintianling deposit.

| XTL-200-01 | Pb | Th | U | 207Pb/206Pb | 207Pb/206Pb | 207Pb/235U | 207Pb/235U | 206Pb/238U | 206Pb/238U | 206Pb/238U | 206Pb/238U |
|---|---|---|---|---|---|---|---|---|---|---|---|
| | ppm | ppm | ppm | Ratio | 1sigma | Ratio | 1sigma | Ratio | 1sigma | Age (Ma) | 1sigma |
| 1 | 35.19141 | 272.5146 | 839.0176 | 0.045944 | 0.000344 | 0.162493 | 0.01102 | 0.025865 | | 164.6139 | 2.161754 |
| 2 | 73.76998 | 983.2053 | 668.4069 | 0.056646 | 0.003653 | 0.191286 | 0.011927 | 0.0254 | 0.000371 | 161.6922 | 2.330072 |
| 4 | 150.3792 | 1478.992 | 2434.117 | 0.053221 | 0.001753 | 0.189809 | 0.006196 | 0.025806 | 0.00026 | 164.248 | 1.636558 |
| 5 | 177.541 | 1613.307 | 3469.185 | 0.050773 | 0.001484 | 0.181493 | 0.005563 | 0.025795 | 0.000287 | 164.1755 | 1.804772 |
| 6 | 89.49963 | 948.6595 | 1368.104 | 0.053941 | 0.002318 | 0.195042 | 0.00915 | 0.025844 | 0.000324 | 164.4834 | 2.034622 |
| 9 | 48.18519 | 463.219 | 754.9348 | 0.056424 | 0.003868 | 0.201489 | 0.013901 | 0.025834 | 0.00037 | 164.4188 | 2.327648 |
| 10 | 262.391 | 3579.414 | 1976.90 | 0.050209 | 0.001792 | 0.181659 | 0.006409 | 0.026129 | 0.000263 | 166.2778 | 1.651446 |
| 11 | 53.20038 | 672.9266 | 595.3193 | 0.048573 | 0.003976 | 0.160214 | 0.012734 | 0.024771 | 0.000368 | 157.7413 | 2.31654 |
| 12 | 58.98816 | 537.3427 | 1241.274 | 0.048787 | 0.002559 | 0.163396 | 0.008134 | 0.024543 | 0.000279 | 156.3039 | 1.756469 |
| 13 | 120.7236 | 1294.406 | 1980.029 | 0.049217 | 0.001852 | 0.166427 | 0.006337 | 0.024362 | 0.000245 | 155.1676 | 1.540812 |
| 14 | 31.13808 | 305.9625 | 501.9875 | 0.052591 | 0.004471 | 0.176232 | 0.014161 | 0.025056 | 0.000422 | 159.5295 | 2.651531 |
| 15 | 42.34564 | 384.2241 | 605.2877 | 0.056603 | 0.003397 | 0.203768 | 0.011987 | 0.026479 | 0.000415 | 168.4726 | 2.608497 |
| 16 | 63.90019 | 833.8722 | 521.9321 | 0.050131 | 0.003748 | 0.173943 | 0.013597 | 0.025474 | 0.000384 | 162.1592 | 2.414481 |
| 17 | 135.3383 | 1800.57 | 860.6884 | 0.047541 | 0.003192 | 0.1716 | 0.011761 | 0.026629 | 0.00037 | 169.4154 | 2.3223 |
| 18 | 21.52203 | 277.3129 | 178.0327 | 0.047665 | 0.01037 | 0.146333 | 0.028933 | 0.024057 | 0.000702 | 153.2453 | 4.416119 |
| Std-Qh-1 | 80.37841388 | 89.97414043 | 240.590879 | 0.074534707 | 0.002579302 | 1.839666506 | 0.060984163 | 0.179116608 | 0.002102619 | 161.146789 | 2.323197 |
| Std-Qh-2 | 74.33384254 | 87.78273142 | 232.6301125 | 0.075225293 | 0.002667461 | 1.860733494 | 0.065194891 | 0.179223392 | 0.002212988 | 160.73057 | 1.295431 |
| Std-Qh-3 | 82.54678906 | 91.79876654 | 248.9876897 | 0.068769 | 0.016213 | 0.213834 | 0.026246 | 0.038179 | 0.012715 | 160.4832123 | 1.292374 |
| Std-Qh-4 | 79.86785431 | 88.90877652 | 230.3231234 | 0.059313 | 0.006757 | 0.204378 | 0.01679 | 0.028723 | 0.003259 | 160.134521 | 1.139287 |

**Table 7.** LA-ICPMS U–Pb zircon dating data of the XTL06 from the Xintianling deposit.

| XTL-06 | Pb | Th | U | 207Pb/206Pb | 207Pb/206Pb | 207Pb/235U | 207Pb/235U | 206Pb/238U | 206Pb/238U | 206Pb/238U | 206Pb/238U |
|---|---|---|---|---|---|---|---|---|---|---|---|
| | ppm | ppm | ppm | Ratio | 1sigma | Ratio | 1sigma | Ratio | 1sigma | Age (Ma) | 1sigma |
| 1 | 49.18345 | 474.7501 | 930.1068 | 0.05013 | 0.003061 | 0.163289 | 0.010028 | 0.023738 | 0.000309 | 151.2364 | 1.942938 |
| 4 | 32.88755 | 337.7602 | 601.4029 | 0.055275 | 0.004295 | 0.176264 | 0.012733 | 0.024043 | 0.000358 | 153.157 | 2.253025 |
| 5 | 118.9204 | 1275.754 | 2140.997 | 0.047771 | 0.00573 | 0.156585 | 0.00573 | 0.023865 | 0.000239 | 152.0385 | 1.505664 |
| 6 | 29.06759 | 305.5499 | 484.7007 | 0.054763 | 0.004053 | 0.17707 | 0.012689 | 0.024097 | 0.000374 | 153.4969 | 2.352972 |
| 7 | 50.5634 | 522.9461 | 939.0106 | 0.042862 | 0.002607 | 0.142702 | 0.008717 | 0.024115 | 0.000277 | 153.6105 | 1.744774 |
| 8 | 43.49626 | 420.021 | 866.6894 | 0.050735 | 0.003087 | 0.169779 | 0.00977 | 0.024627 | 0.000357 | 156.8348 | 2.243882 |
| 9 | 86.25562 | 705.0269 | 1877.608 | 0.05344 | 0.00213 | 0.181933 | 0.007077 | 0.024619 | 0.000253 | 156.7794 | 1.593041 |
| 10 | 87.45636 | 913.5775 | 1588.677 | 0.05087 | 0.002161 | 0.167509 | 0.006758 | 0.023973 | 0.000267 | 152.7182 | 1.677779 |
| 12 | 55.85648 | 556.9221 | 964.2869 | 0.051916 | 0.002559 | 0.175816 | 0.008175 | 0.024972 | 0.000336 | 159.0044 | 2.111661 |
| 13 | 24.13823 | 226.6071 | 445.012 | 0.053609 | 0.004565 | 0.184875 | 0.015556 | 0.025374 | 0.000403 | 161.529 | 2.533204 |
| 14 | 32.09021 | 267.1286 | 706.482 | 0.05031 | 0.003318 | 0.169702 | 0.010109 | 0.024982 | 0.000348 | 159.0658 | 2.188191 |
| 15 | 68.12095 | 661.1316 | 1358.808 | 0.048179 | 0.002254 | 0.159421 | 0.007021 | 0.024178 | 0.000275 | 154.0101 | 1.729854 |
| 16 | 46.29636 | 489.1592 | 875.4446 | 0.053796 | 0.002973 | 0.174047 | 0.00894 | 0.023738 | 0.000268 | 151.2355 | 1.688236 |
| 18 | 72.21149 | 668.9532 | 1304.427 | 0.05003 | 0.002163 | 0.165644 | 0.007092 | 0.024052 | 0.000269 | 153.2138 | 1.6907 |
| Std-TEM | 89.80875908 | 27.34027612 | 905.9055444 | 0.060127027 | 0.001658881 | 0.812591025 | 0.022311531 | 0.097591586 | 0.000757963 | 416.27543 | 2.898142235 |
| Std-TEM | 93.26105582 | 28.00151758 | 912.467429 | 0.062059732 | 0.001737264 | 0.836959751 | 0.02298062 | 0.097579211 | 0.000835196 | 417.98324 | 2.866857765 |
| Std-TEM | 91.36897271 | 26.47448059 | 892.6191825 | 0.063931115 | 0.001966672 | 0.862886258 | 0.026177386 | 0.097579307 | 0.000825982 | 415.89761 | 1.026684593 |

**Table 8.** LA-ICPMS U–Pb zircon dating data of the XTL340 from the Xintianling deposit.

| XTL-340 | Pb ppm | Th | U | 207Pb/206Pb Ratio | 207Pb/206Pb | 207Pb/235U | 207Pb/235U | 206Pb/238U | 206Pb/238U | 206Pb/238U | 206Pb/238U |
|---|---|---|---|---|---|---|---|---|---|---|---|
| | | ppm | ppm | | 1sigma | Ratio | 1sigma | Ratio | 1sigma | Age (Ma) | 1sigma |
| 1 | 162.7584 | 1327.97 | 2786.917 | 0.061992 | 0.00208 | 0.216541 | 0.007261 | 0.025216 | 0.000261 | 160.5397 | 1.639684 |
| 2 | 112.4008 | 804.2761 | 2622.732 | 0.052432 | 0.001854 | 0.184311 | 0.00631 | 0.025386 | 0.000238 | 161.6042 | 1.494933 |
| 3 | 156.5895 | 1286.701 | 3143.719 | 0.050913 | 0.001425 | 0.181325 | 0.005049 | 0.025704 | 0.000236 | 163.6059 | 1.481032 |
| 4 | 289.9565 | 4001.691 | 2070.173 | 0.063124 | 0.001981 | 0.221424 | 0.007151 | 0.025267 | 0.000242 | 160.8568 | 1.522363 |
| 5 | 125.0608 | 983.9794 | 2797.928 | 0.04971 | 0.001495 | 0.174279 | 0.005068 | 0.025412 | 0.000242 | 161.7666 | 1.40774 |
| 6 | 162.3233 | 1400.659 | 3221.655 | 0.050923 | 0.001605 | 0.183729 | 0.00577 | 0.026005 | 0.000212 | 165.4973 | 1.329999 |
| 8 | 126.8413 | 1054.429 | 2488.264 | 0.05176 | 0.001811 | 0.186046 | 0.00659 | 0.026183 | 0.00036 | 166.6115 | 2.261453 |
| 9 | 117.7929 | 909.4558 | 2589.701 | 0.04697 | 0.001462 | 0.164988 | 0.005161 | 0.025367 | 0.000238 | 161.487 | 1.494871 |
| 10 | 185.7832 | 1665.509 | 3407.976 | 0.05224 | 0.001583 | 0.183149 | 0.005569 | 0.025345 | 0.000287 | 161.3463 | 1.805077 |
| 12 | 183.1203 | 1590.276 | 3739.657 | 0.04901 | 0.001424 | 0.171262 | 0.005023 | 0.025193 | 0.000199 | 160.3941 | 1.253863 |
| 13 | 172.2477 | 1509.76 | 3400.123 | 0.051271 | 0.001511 | 0.179143 | 0.00522 | 0.025252 | 0.000211 | 160.7609 | 1.326957 |
| 16 | 401.497 | 5867.655 | 3701.837 | 0.049571 | 0.001439 | 0.170975 | 0.004864 | 0.024963 | 0.000221 | 158.9448 | 1.389775 |
| 18 | 111.6549 | 926.5998 | 2144.36 | 0.052337 | 0.00177 | 0.183989 | 0.006069 | 0.025476 | 0.000214 | 162.1699 | 1.345771 |
| Std-TEM | 94.26076477 | 27.76954828 | 931.7765241 | 0.061328438 | 0.001754657 | 0.833390969 | 0.024089262 | 0.097708883 | 0.000865943 | 417.2342 | 1.910144656 |
| Std-TEM | 95.43700819 | 28.4993559 | 944.7604326 | 0.058840971 | 0.001713827 | 0.808673803 | 0.023492584 | 0.098886659 | 0.000897788 | 415.3452 | 2.294169098 |
| Std-TEM | 94.55323502 | 28.73544114 | 949.1613759 | 0.061328438 | 0.001754657 | 0.833390969 | 0.024089262 | 0.097708883 | 0.000865943 | 419.2334 | 1.560782907 |

The zircons in the three samples were generally 90–150-μm in length, with a length/width ratio of 2–5. It was also found that they were colorless and transparent with some salient microcracks. Moreover, oscillatory zoning was developed in single-grain zircon, with distinct genetic characteristics of magmatic rocks. The Th/U ratios of XTL-200-01-03 and XTL-340-7 were 0.32–2.09 (mean 1.01) and 0.31–1.56 (mean 0.51). Granite porphyry was 0.37–0.63 (mean 0.52).

The testing results of the three samples in this study indicated that the zircon samples are of magmatic origin. The XTL-200-01-03 sample was fresh, medium–coarse biotite granite. The tests revealed that the weighted average 206Pb/238U age was 162.3 ± 1.2 Ma, and MSWD = 1.3 (18 test sites1). The XTL-06 sample was a fresh granite porphyry sample, which was collected from the surface of the gold source ore block in the Xintianling mining area. The weighted average 206Pb/238U age of zircon from 18 sites was 154.3 ± 1.6 Ma, and MSWD = 2.4 (18 test sites). The XTL-340-7 sample was a fresh, fine-grained biotite granite sample, which was collected from the middle section 340 of the Xiaobanlong ore block in the Xintianling mining area. The weighted average 206Pb/238U age of zircon from 18 sites was estimated to be 161.8 ± 1.3 Ma, and MSWD = 1.8, as shown in Figures 5–8.

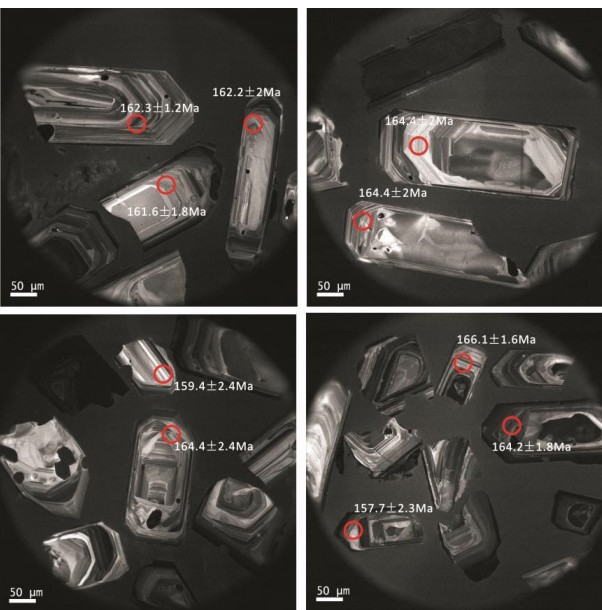

**Figure 5.** CL images of selected dated zircons from medium- and coarse-grained biotite granite. The red circles and nearby numbers indicate the spot locations and concordant ages.

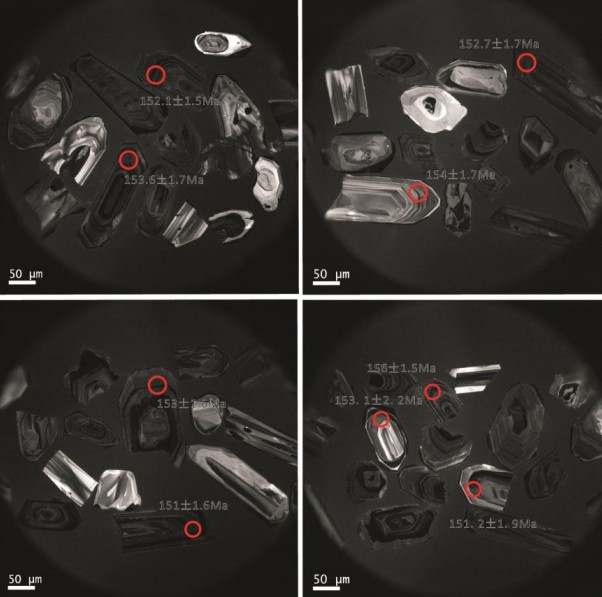

**Figure 6.** CL images of selected dated zircons from granite porphyry. The red circles and nearby numbers indicate the spot locations and concordant ages.

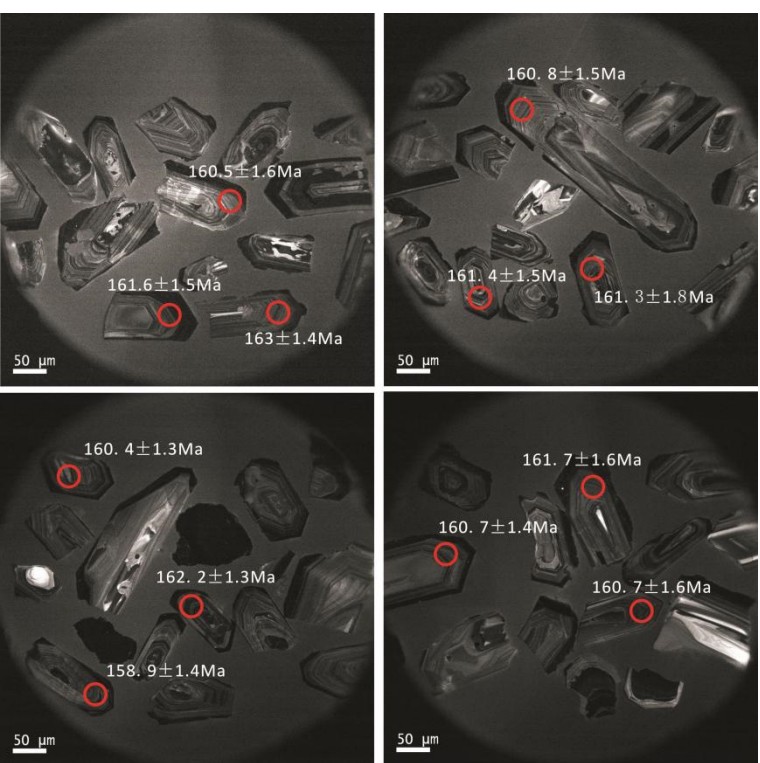

**Figure 7.** CL images of selected dated zircons from fine-grained biotite granite. The red circles and nearby numbers indicate the spot locations and concordant ages.

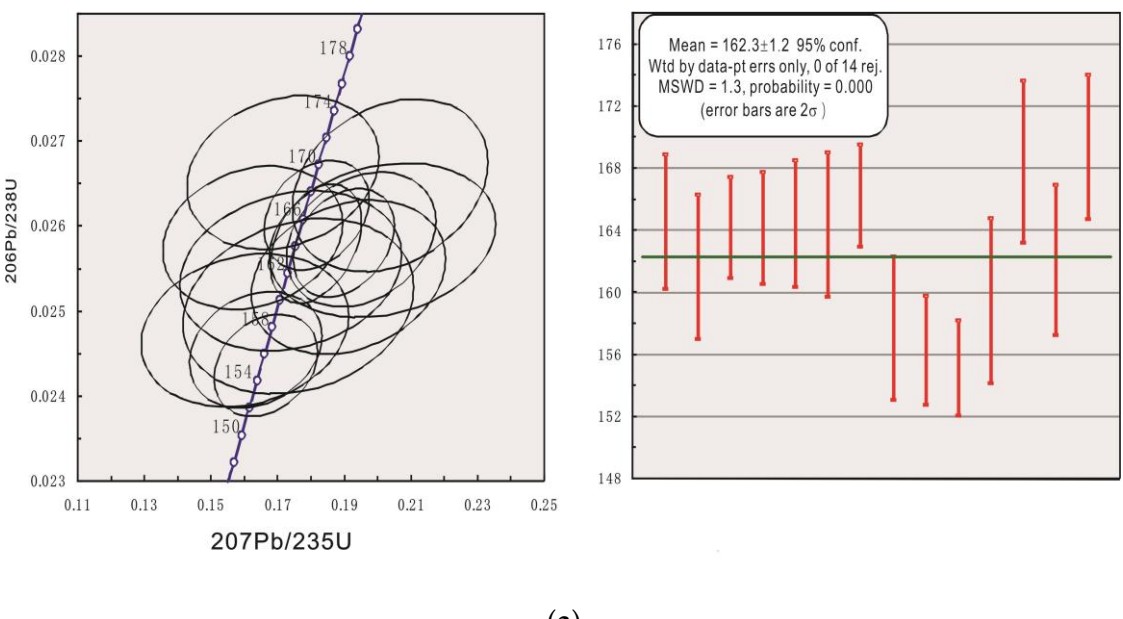

(**a**)

**Figure 8.** *Cont*.

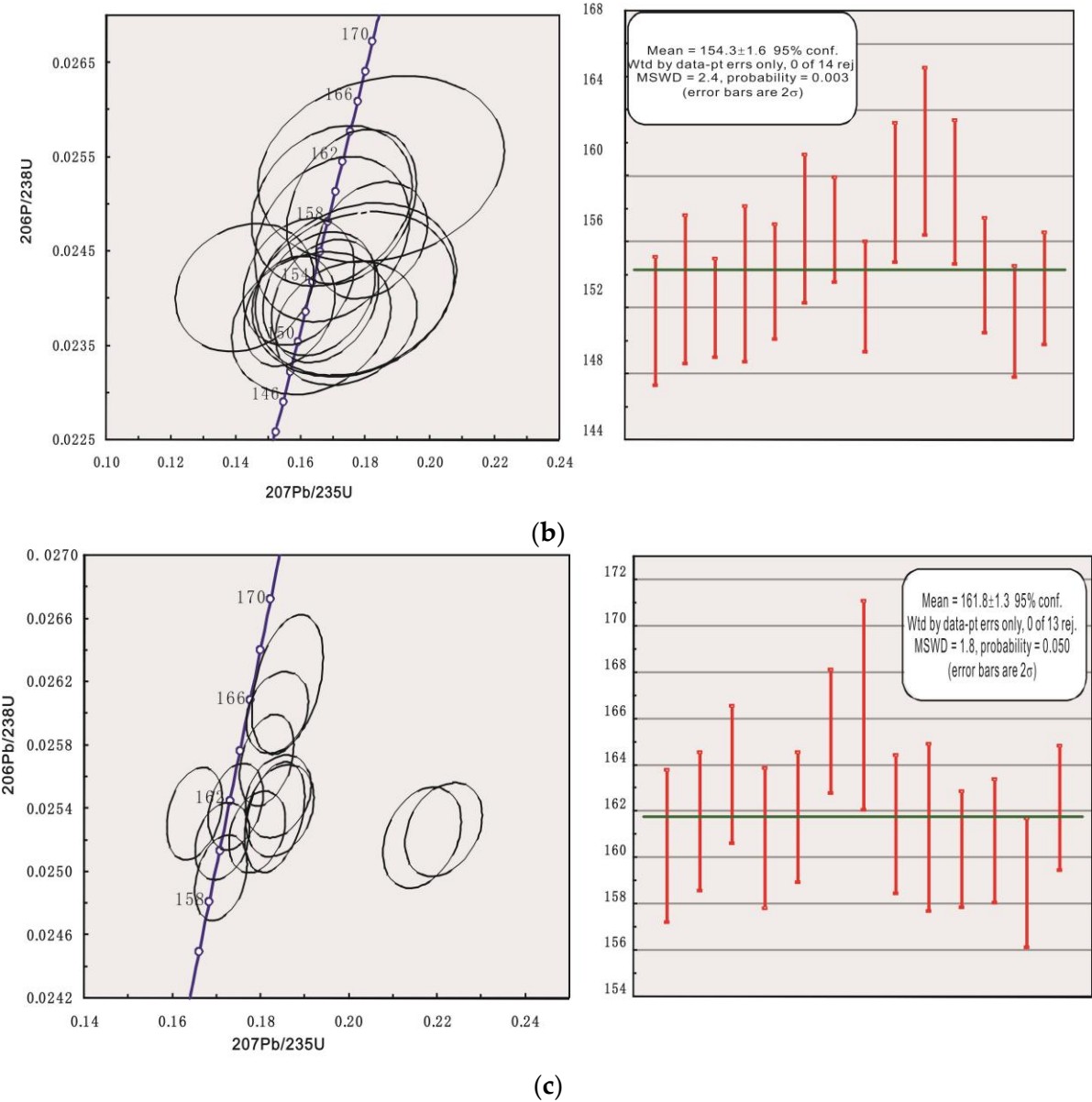

**Figure 8.** Zircon LA-ICP-MS U–Pb concordia diagrams for the samples XTL-200-01 (**a**), XTL-06 (**b**), XTL-340 (**c**).

## 6. Discussion

### 6.1. Data Analysis

The distribution of trace elements and the ratios of elements with similar geochemical characteristics are both sensitive indicators in geological processes, which allow them to be particularly useful in mineral exploration and genesis. Given their highly similar geochemical behaviors, some REEs contain useful information regarding geological evolution. Figure 9 illustrates the REEs in granite in Xintianling, which is normalized to chondrite. The REE contents indicate significant fractioning between LREEs and HREEs, with LREE/HREE ratios of 6.22 to 12.33 and $(La/Yb)_N$ ratios of 6.65 to 24.78. Delta Eu ranged from 0.09 to 0.67, with an average value of 0.52, signifying a distinct negative anomaly. In contrast, δCe (0.93 to 1.06) exhibited no conspicuous anomalies. The enrichment in LREE, the relative depletion of HREE, and a negative Eu anomaly all formed a V-type pattern (Figure 9). The consistent REE patterns indicate identical or similar sources and evolution processes. Figure 10 shows the ratio spider diagram of trace elements in the analyzed samples. The rocks depleted Ba, Ti, and Sr contents and enriched Hf and U contents. The depletion in

Ba, Ti, and Sr contents (normalized to primitive mantle) suggests that fractional crystallization was essential in their petrogenesis. Moreover, the rocks exhibited characteristics of non-orogenic granites.

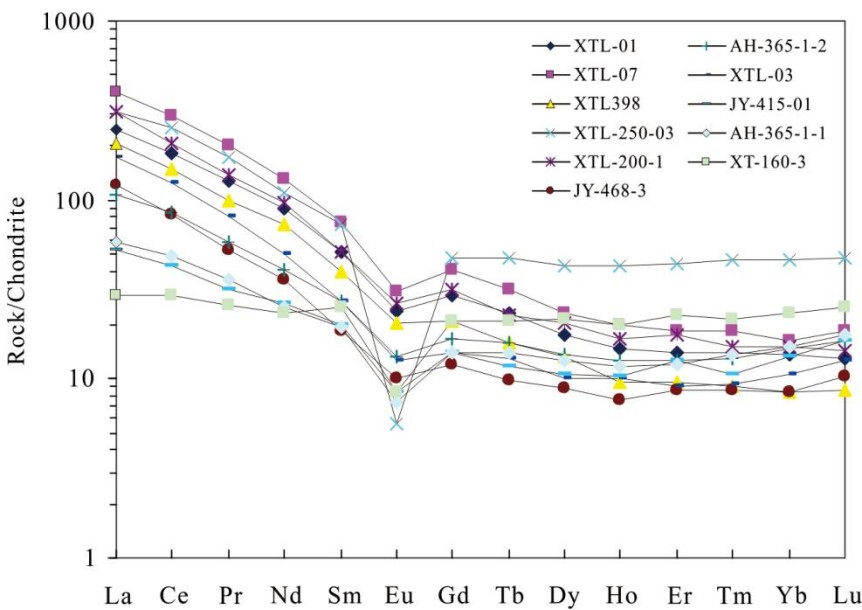

**Figure 9.** Chondrite-normalized REE patterns for granites in Xintianling ore field, Hunan, China.

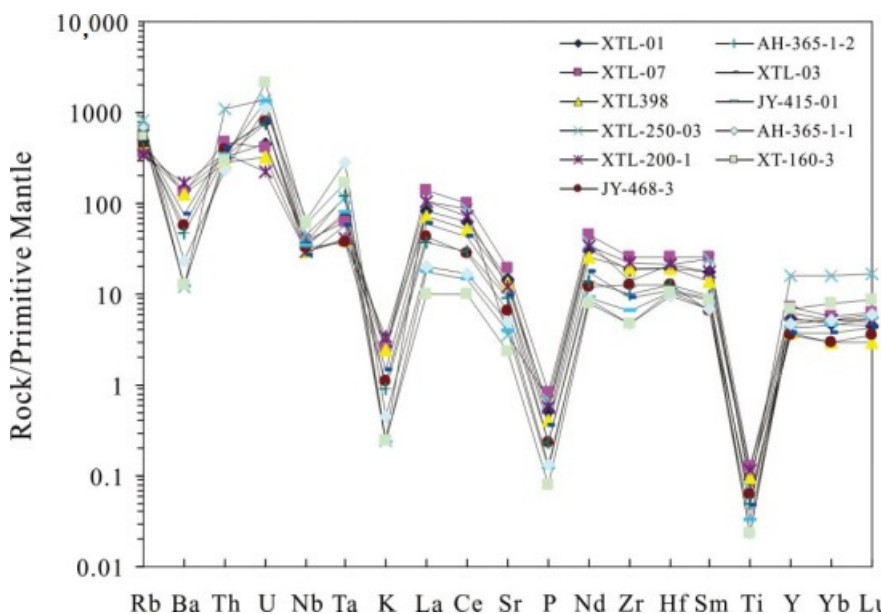

**Figure 10.** Primitive mantle-normalized spider diagram of trace elements for granites in Xintianling ore field, Hunan, China.

### 6.2. Genesis of Granite

Regionally, tin mineralization occurs mostly in the west and is related to intracontinental granite, whereas tungsten mineralization mainly occurs in the east and is related to continental margin granite. The two mineralizations are formed along with the tectonic evolution of diagenesis and the mineralization of the Nanling metallogenic belt.W-Sn metallogenic attributes, related to both intracontinental and epicontinental granite in southern Hunan, have been identified. The mineralization of tungsten and tin was previously considered to be strongly associated with Yanshanian S-type granite (such as Yaogangxian, Dengfuxian, Xihuashan, Dajishan, Tianmenshan, Hongtaoling, and Tieshanlong). Of

these types of granite, some represent transitional types, from I-type to S-type and a small amount of A-type granite. However, scholars have recently reported that many types of granite in the Nanling region are related to tin and tungsten–tin mineralization (such as Qitianling, Huashan-Guposhan, Laiziling, Jiuyishan, Xitian, Qianlishan, etc.) and represent A-type granite. In this context, Xintianling granite can be divided into three types: main-body granite, complement granite, and vein rock granite. Main granite is primarily biotite granite, and dike is primarily granite porphyry and quartz porphyry, which cuts through the early-emplaced rock mass. It is rich in aluminous biotite and formed by the anatexis of lower dry crustal materials, which are affected by the underplating or injection of mantle-derived magma, resembling S-type granite. However, the characteristics of mineral cordierite were not elucidated in this study, and the formation temperature was similar to that of granite porphyry (743 °C). Notably, our results indicated that biotite granite, related to mineralization in the Xintianling tungsten deposit, is A-type granite. (Figures 11 and 12).The following evidence of this pattern was demonstrated:

(1) The biotite granite was rich in Si and K but poor in Mg and Ca while being peraluminous.
(2) The biotite granite featured high Zr and Rb levels and low Sr and Ba levels. It was enriched by Rb, U, Ta, and Pb, but levels of Ba, K, Nb, La, and Ce were depleted.
(3) In terms of REEs, light REEs enriched the granite. There was a strongly negative anomaly regarding europium, and a V-shape REE distribution pattern was formed.
(4) The diagram of ($Na_2O + K_2O$/CaO-($Zr + Nb + Ce + Y$)) indicated that most of the biotite granite was in the area where A-type granite was located. Likewise, the diagram of ($Na_2O + K_2O$)/CaO—10,000 Ga/AL suggested that most of the biotite granite was within the area where the A-type granite was located, while a small number of samples were within the I or S-type granite location, and granite porphyry was either within or near the I or S-type granite locations.

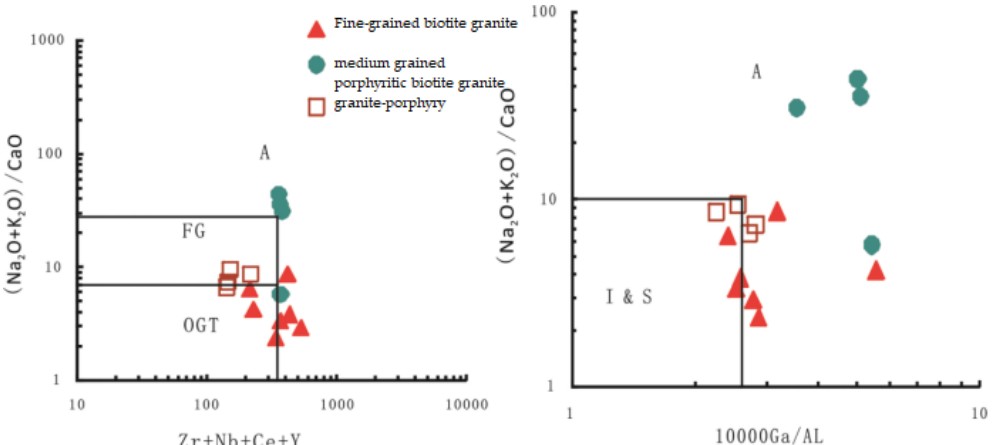

**Figure 11.** Plots of ($Na_2O + K_2O$)/CaO versus ($Zr + Nb + Ce + Y$) and ($Na_2O + K_2O$)/CaO versus 10,000 Ga/AL of the Xintianling granite. The areas indicated in the above diagrams are from J.B. Whalen [26].

### 6.3. Geodynamic Settings

The geodynamic background and large-scale mineralization of tungsten and tin deposits in South China have always attracted substantial attention from geologists. Specifically, large-scale W-Sn polymetallic deposits in South China are thought to have formed catastrophically in the Yanshanian period of 160 to 150 Ma. The formation of these deposits was closely related to granite. The current understanding of the tectonic setting of the Jurassic granite in South China is the result of multiple research efforts conducted from different perspectives. One of the most controversial issues in this research field is whether the subduction of the paleo-Pacific plate occurred in South China in the early Yanshanian period or whether this was simply intraplate tectonism. Intraplate extensional

decompression or intracontinental subduction, compression, thickening, and redissolution were suggested to have occurred.

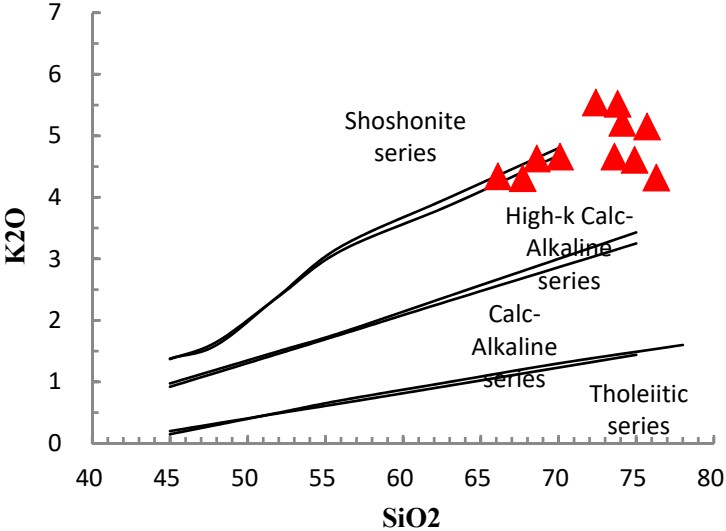

**Figure 12.** Granite-class diagram of $SiO_2$–$K_2O$ (Peccerillo [28]).

Within the literature regarding the Nanling W-Sn metallogenic belt, early scholars suggested that the belt was related to S-type granite, which was remelted by the continental crust through collision. However, the discovery of an NE-trending, alkali-rich intrusive rock belt in Qianlishan–Qitianling–Xishan–Huashan and Guposhan rock mass structures in the middle of the Nanling mountain range has created the opportunity for another working hypothesis to be researched. Other scholars have suggested that exposed granite primarily represents S-type granite. This granite was formed in a post-orogenic extensional setting, rather than through collisional compression.

There is a general relationship between types of granite and tectonic environments. Granite formed in different tectonic settings commonly exhibits distinct chemical composition, P-T conditions, and formation mechanisms. In turn, these characteristics can be used to determine the tectonic environment of granite, and they also lay the foundation for the study of regional tectonic evolution. Trace elements in rocks can be used to determine their tectonic environment. The rock trace-element ratios are plotted as (Y + Nb) − Rb, (Yb + Ta) − Rb, Yb − Ta, and Y − Nb in Figure 13. As indicated by the tectonic environment discrimination diagram of (Y + Nb) − Rb, five rock samples were in the syn-COLG area; four were in the WPG area. Furthermore, Figure 13, with (Yb + Ta) − Rb, indicates that seven rock samples were found in the syn−COLG area, and the others were found in the WPG area. Figure 13, which illustrates Yb − Ta, demonstrates that most samples were in the WPG area, while Figure 13, which displays Y − Nb, reveals that most samples were in the VAG + syn−COLG area.

Overall, the trace-element compositions of the Xintianling intrusion suggest that it was affiliated with the Tethys-collision when the tectonic setting was transitioned from collisional orogeny/within-plate tectonism to intraplate collision orogeny.

Late Yanshanian tectonic conversion is an important driver of Xintianling magmatic activities and also provided conductive dynamic conditions for the large-scale mineralization that occurred in South China. The magmatic activities are governed by the tectonic environment. As such, their chemical compositions feature a genetic relationship with the tectonic environment. The Gd/Lu of granite ranged from 15.00 to 20.00 in extensional settings and from 8.00 to 12.00 in compressive settings. The Gd/Lu of Xintianling granite, ranging between 11.54 and 16.77, was generally higher than granite in the compressive environment. However, it was close to that of granite in the extensional environment, potentially indicating that magma intruded in extensional settings. The Nanling metallogenic belt within the main ore-forming tectonic environment is a relatively stable regional tension.

The emplacement age represents a transitional period for the tectonic environment, namely, the transition from the Indosinian orogenic to the late intraplate extensional period, which occurred in the late Yanshanian era. More specifically, the crustal deformation and magmatic activity during the Indosinian–Yanshanian period of ore-forming materials promoted the occurrence and development of mineralization, thereby creating favorable conditions for large-scale mineralization [27,28].

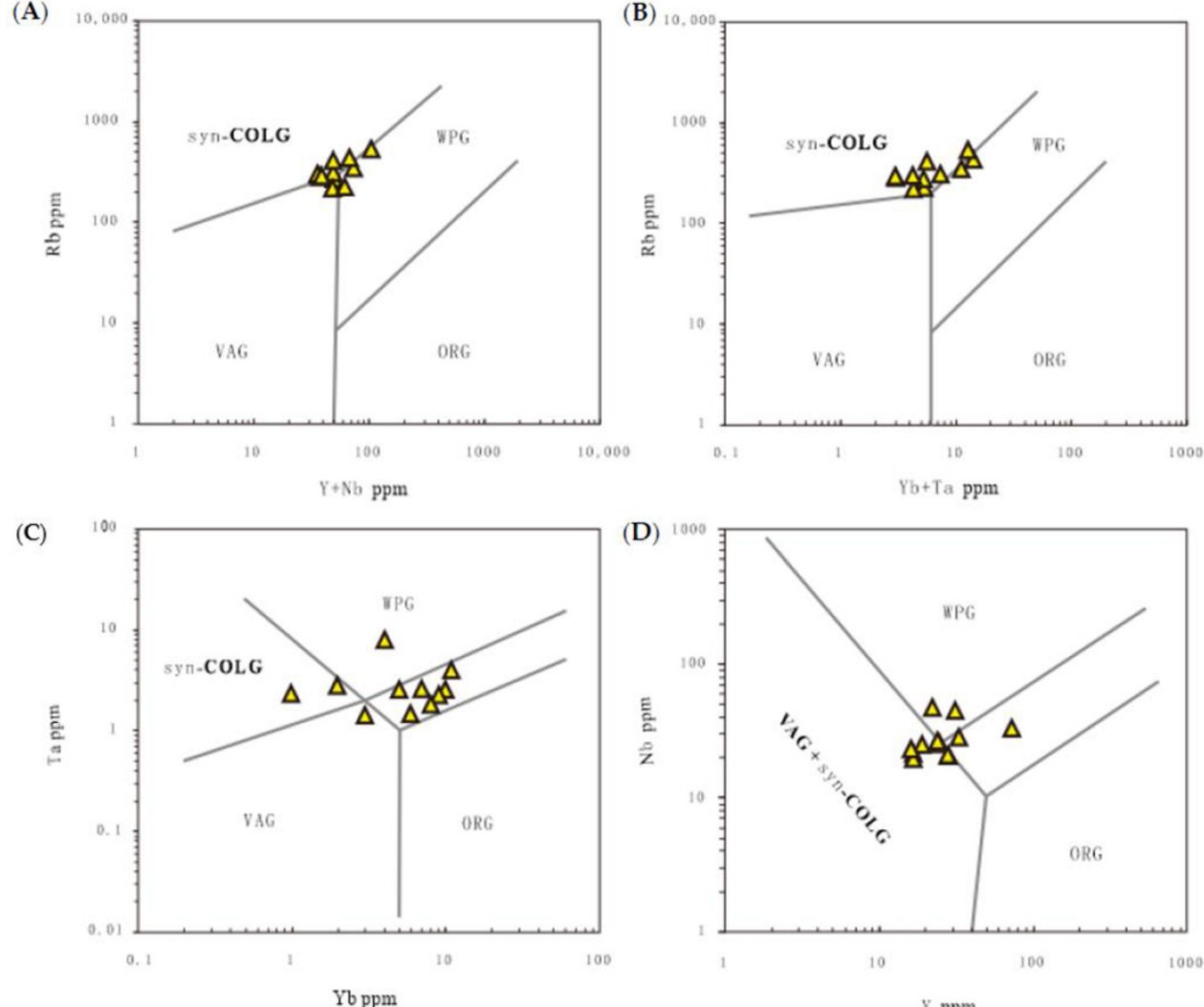

**Figure 13.** (**A**) Rb − (Y + Nb), (**B**) Rb − (Yb + Ta), (**C**) Ta − Yband (**D**) Nb − Y discrimination diagrams of granites (Pearce J A, [27]) VAG—volcanic arc granite; syn-COLG—syn collisional granite; WPG—within plate granite; ORG—ocean ridge granite.

## 7. Conclusions

This study examined the regional geological characteristics, deposit geological characteristics, and elemental geochemical characteristics of the Xintianling tungsten deposit. The analysis revealed that the deposit presented a typical skarn-type scheelite. It was the product of large-scale mineralization in Nanling in the middle–late Jurassic era (160–150 Ma). The results indicated that lithospheric extension, asthenosphere upwelling along deep faults, and intensive mantle–crust interaction processes all likely occurred under the tectonic setting in the Mesozoic era.

(1) The average $SiO_2$ content levels in Xintianling granite were estimated to be 72.11 wt%. $K_2O + Na_2O > 7.00\%$, i.e., the level of $w(K_2O)/w(Na_2O)$ was high. The $Al_2O_3$

content levels were generally high, with an average of 13.49 wt%. Overall, the studied granite was found to belong to the high-potassium calc-alkaline series.

(2) The fractionating degrees of light and heavy REEs were higher, and the chondrite standardized distribution pattern coincided well with these. This indicates similar magma sources and evolution histories.

(3) The evolution of granite intrusions was affected by crust–mantle interaction, which supplied the material source.

(4) The rock was likely to have formed from partial melting by means of crust–mantle interactions in the period of the collision orogeny and intraplate. This pattern was peculiarly prominent during the conversion period of both in the tensile tectonic environment.

(5) LA-ICP-MS zircon U-Pb dating indicated the middle–coarse biotite granite (162.3 ± 1.2 Ma, MSWD = 1.3), the fine-grained biotite granite (161.8 ± 1.3 Ma, MSWD = 1.8), and the granite porphyry (154.3 ± 1.6 Ma, MSWD = 2.4) were formed in the late Jurassic period. The scheelite was revealed to have an Sm-Nd isochron age of 157.1 ± 3.2, which is within the margin of error of molybdenite's Re-Os isochron age of 159.1 ± 2.6 Ma. Both ages agreed well with the zircon U-Pb ages, thus suggesting that the formation of the Xintianling scheelite deposit was synchronous with the development of granite intrusions and that ore-forming fluids were derived from these intrusions.

**Author Contributions:** Conceptualization, W.Y.; methodology, W.Y.; software, M.Z.; formal analysis, W.Y.; investigation, M.Z. and W.Y.; data curation, J.Y.; writing—original draft preparation, W.Y.; writing—review and editing, X.C. and W.Y.; visualization, J.Y.; supervision, J.Y. All authors have read and agreed to the published version of the manuscript.

**Funding:** This research was supported by the Major Research plan of National Science Foundation of China [92062106], the Science and Technology Support Plan of Guizhou Province [Qian (2017)1078] and Qian (2019)1138], the High-level talent introduction program for the Guizhou Institute of Technology (0203001018040).

**Institutional Review Board Statement:** Not applicable.

**Informed Consent Statement:** Not applicable.

**Data Availability Statement:** Not applicable.

**Conflicts of Interest:** The authors declare no conflict of interest.

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
