# Peer review of "Zircon U-Pb Ages and Geochemistry of the Granite in the Xintianling Tungsten Deposit, SE China: Implications for Geodynamic Settings of the Regional Tungsten Mineralization"

_minerals, doi:10.3390/min12080952_

Round 1

Reviewer 1 Report

The paper proposed by the Authors deals with the petrological features and geochronology of a granite intrusion related to W mineralizing events in the Nanling district (SE China). The topic certainly falls within the scientific interests of Minerals, however, during the review of the work I encountered a series of critical issues that do not allow us to consider its publication in the present form. In particular:

1)      despite the title, the work is almost totally focused on the petrological study through whole-rock geochemistry and geochronological determinations; there is no adequate description of the mineralized systems (Chapter 2), nor an adequate discussion on the metallogenic implications of the study carried out (Chapter 6);

2)      entire parts of the text require a profound restructuring: the best example is the Conclusions paragraph, which appears totally incoherent and disjointed from the text;

3)       References are a total mess, and the reference list is not complete;

4)      Many problems derive from a largely inadequate English language that makes some sentences confusing and incomprehensible;

5)       Further and numerous observations are present in the attached text.

In summary, at present the paper cannot be accepted in its current form; I recommend the Authors to revise it in depth, also for English, and to integrate the missing parts.

Author Response

Dear reviewer:

Thank you very much for your guidance and advice. We have learned much from your and two other reviewers’ comments, which are fair, encouraging and constructive. After carefully studying the comments and your advice, we have made corresponding changes. The main revisions are listed below.

(1)According to the comments and contents of article,The title of the article has been amended.We have changed a scientific and reasonable titles: Zircon U-Pb Ages and Geochemistry of the Granite in the Xintianling Tungsten Deposit, SE China: Implications on geodynamic settings of the regional tungsten minerlization.

(2)The Abstract section was rewritten to highlight the conclusions as suggested by the reviewers.

(3)According to the reviewer's request, Regional Geological Characteristics was rewritten,Corresponding to the topic, the regional structure is mainly introduced

(4)The Discussion section has been rearranged in response to scientific issues raised in Introduction. 

(5)Supplement the lack of standardization and standard sample part of the test

(6)References were renumbered and optimized.Due to forgetting to adjust the reference when revising the article(Deleted a lot of content.), there was confusion in the reference labeling, which has been corrected.

(7)We supplement the original U-Pb zircon data and discard the Concordia map of U-Pb which have problems 

(8)We have polished the language with the help of MDPI official.

(9)Some figures have been rearranged to avoid contradiction. Some approaches have been reconsidered to make estimation and calculation more convinced and accurate.

(10)Some contents have been cut to make the manuscript more concise, as suggested by the reviewers.

Response to Reviewer 1(Please see the attachment for modification remarks)

1.Carefully revise the abstract for English, as several statements appear confusing

Revision:According to your approval and suggestion, we have revised the abstract.

The Xintianling tungsten deposit is a super-large deposit in the Nanling tungsten-tin mineralization belt(China),which is genetically associated with the early stage hornblende-biotite monzonitic granite of Qitianling pluton. The orebodies predominant occur as veins and lenses within skarn rocks between Xintianling granite and limestone(Shidengzi group). In this work, whole-rock major and trace elements and zircon U–Pb ages of the Xintianling granite were studied in an attempt to investigate the geochronological framework, petrogenesis, tectonism, and metallogenesis with regard to the deposit. The petrographic and geochemical analyses indicated that the Xintianling granite consists of three intrusive unites of medium- and coarse-grained biotite granite, fine-grained biotite granite, and granite porphyry, of which, the biotite granite was strongly associated with mineralization. Biotite granite rocks are highly K-calc-alkaline and weakly peraluminous, with A/CNK ratios ranging from 0.99 to 1.05. Late granite porphyry is aluminum-supersaturated with a high evolution degree, whose geochemical characteristics suggest that it is either an I- or S-type granite. LA-ICP-MS zircon U-Pb dating revealed that medium- and coarse-grained biotite granite (162.3±1.2 Ma, MSWD = 1.3), fine-grained biotite granite (161.8±1.3 Ma, MSWD = 1.8), and granite porphyry (154.3 ± 1.6 Ma, MSWD = 2.4) formed in the late Jurassic. The emplacement of the Qitianling A-type granite and associated tungsten-tin polymetallic mineralization is a continuous evolution process,which are the products of large-scale mineralization of the Nanling in Middle-Late Jurassic (150-160Ma).Under the tectonic setting of Mesozoic lithospheric extension,asthenosphere upwelling along deep fault,intensive mantle-crust interaction processes probably provide not only the high heat flow,but also part mantle-derived material for large-scale W-Sn-polymetallic mineralization in this area. 

2.Control your references, I have the impression that there is something out of place.

Revision :I'm really sorry for this mistake. We have carefully checked all references and renumbered them 

3.? what do you mean? the term diagenesis is more related to sedimentary environments.

Revision: :rewrote the paragraph

To clarify the above ambiguity, we focus on the tungsten-related granite in the Xintianling deposit in south China. We first presented major and trace element compositions of the granite to clarify its genesis. Then the zircon U-Pb dating was conducted to place geochronologic constraints on intrusions and related mineralization. Our new results shown here, when combined with other independent evidence, demonstrate that the Xintianling tungsten deposit and regional mineralization formed in late Jurassic under a back-arc extensional setting. The events were likely triggered by the break-off or detachment of the flatly subducted Paleo-Pacific slab

  1. Not clear, explain this structural setting with reference to Figure 1 and  What rocks these units are made of? modify this stratigraphy

Revision: we have changed the Figure and description.

Figure 2 Tectonic setting of the Qianlishan-Qitianling area [7].

5.These descriptions are quite simplistic and MUST be redone ad improved, although schematically, in terms of modern ore deposits descriptions, if you want to correctly highlight the relationships between granite petrology and ore deposits.  You may  refer to

Zhang, Rongqing & Lu, Jianjun & Huang, Xudong & Zhang, Qiang & Zhao, Xu & Li, Xiaoyu. (2019). Formation of the Xintianling scheelite skarn deposit, Nanling Range, South China: insights from petrology, mineral chemistry and C-H-O-S-Pb isotopes. : Formation of the Xintianling scheelite skarn deposit, Nanling Range, South China: insights from petrology, mineral chemistry and C-H-O-S-Pb isotopes.  in: Life with Ore Deposits on Earth – 15th SGA Biennial Meeting 2019, Volume 1, 186-189. and, for general features, to Meinert et al (2005) Meinert L.D., Dipple G.M. and Nicolescu S. (2005) World Skarn Deposits, Soc. Econ.  Geol. , Economic Geology 100th Anniversary Volume, 299-336, DOI: 10.5382/AV100.11.

Revision: :rewrote the paragraph

The Xintianling deposit is located north of the Qitianling pluton, which represents the central part of the Nanling Range . Strata are primarily carboniferous in age, which are covered locally by quaternary sediments. The carboniferous strata, from young to old, can be classified into the lower carboniferous YanGuan stage, the DaTang stage, and the middle-early Carboniferous TianHu stage. The principal structure of this area is in north–south, east–north, east–west directions. Its mineralization is related to a north–south Duplex anticline. The main body of granite is micaceous granite, while the grain size of the main body of granite rock decreases from center to edge. In turn, it can be divided into three lithologies: coarse-grained porphyritic biotite granite, medium- and coarse-grained porphyritic biotite granite, and fine–medium- and coarse-grained porphyritic biotite granite.

The main ore minerals present in the Xintianling deposit are scheelite and molybdenite, followed by small amounts of bismuthinite, galena, sphalerite, chalcopyrite, arsenopyrite, pyrrhotite, and pyrite. The main gangue minerals are garnet, diopside, actinolite, iron mica, chlorite, quartz, and small amounts of calcite, fluorite, epidote, etc. The main granular textures of the ores located here are automorphic, hypidiomorphic, and xenomorphic, as well as vein rocks. It should be noted that the main ore structures are disseminated structures. The main wall rock alterations that occur are skarnization silicification, greisenization, and marmarization. Notably, skarnization exhibits the most correlation with metallogenesis.

  1. despite the title, the work is almost totally focused on the petrological study through whole-rock geochemistry and geochronological determinations; there is no adequate description of the mineralized systems (Chapter 2), nor an adequate discussion on the metallogenic implications of the study carried out (Chapter 6);…

Revision: Indeed, this is a mistake in our working, according to your suggestion and the content of the article   We have changed a scientific and reasonable titles: Zircon U-Pb Ages and Geochemistry of the Granite in the Xintianling Tungsten Deposit, SE China: Implications on geodynamic settings of the regional tungsten minerlization.A lot of adjustments have been made to the chapter contents. Please see the attachment for details

7.entire parts of the text require a profound restructuring: the best example is the Conclusions paragraph, which appears totally incoherent and disjointed from the text;./ Many problems derive from a largely inadequate English language that makes some sentences confusing and incomprehensible; (There are many supplementary and modified parts, please refer to the attachment)

Revision: .A lot of adjustments have been made to the chapter contents. Please see the attachment for details.We have polished the language with the help of MDPI official.

8.References are a total mess, and the reference list is not complete;.

Revision: References were renumbered and optimized.I'm very sorry. We have carefully checked and revised all the references

More detailed revisions are annotated in the revised manuscript. (As required, I used the modification mode in Word)

Reviewer 2 Report

the manuscript has potential. The data are there and can definitely end up in a publication. Unfortunately this is not yet the case.

needs to be fixed:

1- the manuscript needs to be revised for the English; not the grammar, only the tenses are a major grammar issue in the manuscript, but the structure of all the sentences needs to be revised. 

As it is, it sounds like the authors translated directly from their head into a stream of consciousness with no rule, asking the reader to understand what they mean (I guess Joyce is not one of the co-authors), making the manuscript very hard to understand and somehow/sometimes irritating.

2- data control. there is no mention of any data control, no std for your dissolution trace elements acquisition or U-Pb acquisition.

bear this in mind for any future analytical-based future publications, no std=no data.

3- the way geochron data are interpreted, makes clear to the reader that the authors have either never done anything with geochron data, or let someone else doing this without double-checking before submission, or both.

highly recommend to ask for the help of a geochron person and provide a proper interpretation of those data, the one in the manuscript are wrong, which makes the discussions and conclusions incorrect.

once these points/flaws have been addressed then it can be taken in consideration for publication.

Author Response

Dear reviewer:

Thank you very much for your guidance and advice. We have learned much from your and other two reviewers’ comments, which are fair, encouraging and constructive. After carefully studying the comments and your advice, we have made corresponding changes. The main revisions are listed below.

(1)According to the comments and contents of article,The title of the article has been amended.We have changed a scientific and reasonable titles: Zircon U-Pb Ages and Geochemistry of the Granite in the Xintianling Tungsten Deposit, SE China: Implications on geodynamic settings of the regional tungsten minerlization.

(2)The Abstract section was rewritten to highlight the conclusions as suggested by the reviewers.

(3)According to the reviewer's request, Regional Geological Characteristics was rewritten,Corresponding to the topic, the regional structure is mainly introduced

(4)The Discussion section has been rearranged in response to scientific issues raised in Introduction. 

(5)Supplement the lack of standardization and standard sample part of the test

(6)References were renumbered and optimized.Due to forgetting to adjust the reference when revising the article(Deleted a lot of content.), there was confusion in the reference labeling, which has been corrected.

(7)We supplement the original U-Pb zircon data and discard the Concordia map of U-Pb which have problems 

(8)We have polished the language with the help of MDPI official.

(9)Some figures have been rearranged to avoid contradiction. Some approaches have been reconsidered to make estimation and calculation more convinced and accurate.

(10)Some contents have been cut to make the manuscript more concise, as suggested by the reviewers.

More detailed revisions are annotated in the revised manuscript. (As required, I used the modification mode in Word)

Response to Reviewer 2(Please see the attachment for modification remarks)

  1. 1- the manuscript needs to be revised for the English; not the grammar, only the tenses are a major grammar issue in the manuscript, but the structure of all the sentences needs to be revised. 

Revision: According to the problems pointed out in the PDF feedback from reviewers, we made a lot of modifications.We have polished the language with the help of MDPI official.Please see the attachment for modification remarks

  1. data control. there is no mention of any data control, no std for your dissolution trace elements acquisition or U-Pb acquisition..

Revision: I'm really sorry for such a problem. We have made a supplement in the article 

P6:After petrographic examination and removal of altered surfaces, the samples for wholerock analysis were crushed in an agate mill to 200 mesh. X-ray fluorescence (XRF; Rigaku RIX 2100 spectrometer) using fused glass disks and ICP-MS (Agilent 7500a with a shield torch) were used to measure the major and trace element compositions,respectively, ALS Minerals (ALS Chemex) after acid digestion of samples in Teflon bombs. The analytical

precision was better than 5% for major elements and often better than 10% for trace

elements [21]. The detailed analytical procedures for major element analysis by XRF are

described in [22], while those for trace element analysis by ICP-MS are described in [23].

The results of the analyses for major and trace elements are listed in Table 1-5.

  P10 Zircons were isolated from Medium-grained porphyritic biotite granite and Fine-grained biotite granite using combined magnetic and heavy liquid separation techniques at the Geological Laboratory of the Regional Geological Survey, Langfang City, Hebei Province, China. The zircon grains were examined under transmitted and reflflected light using an optical microscope. Distinct domains within the zircons were selected for analysis based on their CL images. An Agilent 7500a ICP-MS equipped with a 193 nm laser was used. In the experiment, high-purity He was used as the carrier gas of the ablated substance, the laser operating frequency was 10 Hz, the laser spot diameter at the test point was 36 µm, and the effective acquisition time of the mass spectrometer was 45 s. U–Pb isotope fractionation uses the international standard zircon 91500 as the external correction and TEM (416±5Ma) and QH (160 ± 1Ma) as monitoring standards. Samples housed at the State Key Laboratory of Geological Processes and Mineral Resources, China University of Geosciences (Wuhan) was

used to measure the U–Pb ages of zircons. The ICP-MS DataCal (Ver. 6.7) [26] and Isoplot (Ver. 3.0) [27] programs were used for data reduction. The correction for common Pb was made following Anderson (2002). The dating results are presented in Table 6-8.

  1. 3- the way geochron data are interpreted, makes clear to the reader that the authors have either never done anything with geochron data, or let someone else doing this without double-checking before submission, or both.highly recommend to ask for the help of a geochron person and provide a proper interpretation of those data, the one in the manuscript are wrong, which makes the discussions and conclusions incorrect.

Revision: We checked the data, deleted the wrong map, and provided the original test data 

Pb

Th

U

207Pb/206Pb

207Pb/206Pb

207Pb/235U

207Pb/235U

206Pb/238U

206Pb/238U

206Pb/238U

206Pb/238U

XTL-200-01

ppm

ppm

ppm

Ratio

1sigma

Ratio

1sigma

Ratio

1sigma

Age (Ma)

1sigma

1

35.19141

272.5146

839.0176

0.045944

0.000344

0.162493

0.01102

0.025865

164.6139

2.161754

2

73.76998

983.2053

668.4069

0.056646

0.003653

0.191286

0.011927

0.0254

0.000371

161.6922

2.330072

4

150.3792

1478.992

2434.117

0.053221

0.001753

0.189809

0.006196

0.025806

0.00026

164.248

1.636558

5

177.541

1613.307

3469.185

0.050773

0.001484

0.181493

0.005563

0.025795

0.000287

164.1755

1.804772

6

89.49963

948.6595

1368.104

0.053941

0.002318

0.195042

0.00915

0.025844

0.000324

164.4834

2.034622

9

48.18519

463.219

754.9348

0.056424

0.003868

0.201489

0.013901

0.025834

0.00037

164.4188

2.327648

10

262.391

3579.414

1976.90

0.050209

0.001792

0.181659

0.006409

0.026129

0.000263

166.2778

1.651446

11

53.20038

672.9266

595.3193

0.048573

0.003976

0.160214

0.012734

0.024771

0.000368

157.7413

2.31654

12

58.98816

537.3427

1241.274

0.048787

0.002503

0.163396

0.008134

0.024543

0.000279

156.3039

1.756469

13

120.7236

1294.406

1980.029

0.049217

0.001852

0.166427

0.006337

0.024362

0.000245

155.1676

1.540812

14

31.13808

305.9625

501.9875

0.052591

0.004471

0.176232

0.014161

0.025056

0.000422

159.5295

2.651531

15

42.34564

384.2241

605.2877

0.056603

0.003397

0.203768

0.011987

0.026479

0.000415

168.4726

2.608497

16

63.90019

833.8722

521.9321

0.050131

0.003748

0.173943

0.013597

0.025474

0.000384

162.1592

2.414481

17

135.3383

1800.57

860.6884

0.047541

0.003192

0.1716

0.011761

0.026629

0.00037

169.4154

2.3223

18

21.52203

277.3129

178.0327

0.047665

0.01037

0.146333

0.028933

0.024057

0.000702

153.2453

4.416119

Pb

Th

U

207Pb/206Pb

207Pb/206Pb

207Pb/235U

207Pb/235U

206Pb/238U

206Pb/238U

206Pb/238U

206Pb/238U

XTL-06

ppm

ppm

ppm

Ratio

1sigma

Ratio

1sigma

Ratio

1sigma

Age (Ma)

1sigma

1

49.18345

474.7501

930.1068

0.05013

0.003061

0.163289

0.010028

0.023738

0.000309

151.2364

1.942938

4

32.88755

337.7602

601.4029

0.055275

0.004295

0.176264

0.012733

0.024043

0.000358

153.157

2.253025

5

118.9204

1275.754

2140.997

0.047771

0.001854

0.156585

0.00573

0.023865

0.000239

152.0385

1.505664

6

29.06759

305.5499

484.7007

0.054763

0.004053

0.17707

0.012689

0.024097

0.000374

153.4969

2.352972

7

50.5634

522.9461

939.0106

0.042862

0.002607

0.142702

0.008717

0.024115

0.000277

153.6105

1.744774

8

43.49626

420.021

866.6894

0.050735

0.003087

0.169779

0.00977

0.024627

0.000357

156.8348

2.243882

9

86.25562

705.0269

1877.608

0.05344

0.00213

0.181933

0.007077

0.024619

0.000253

156.7794

1.593041

10

87.45636

913.5775

1588.677

0.05087

0.002161

0.167509

0.006758

0.023973

0.000267

152.7182

1.677779

12

55.85148

556.9221

964.2869

0.051916

0.002559

0.175816

0.008175

0.024972

0.000336

159.0044

2.111661

13

24.13823

226.6071

445.012

0.053609

0.004565

0.184875

0.015556

0.025374

0.000403

161.529

2.533204

14

32.09021

267.1286

706.482

0.05031

0.003318

0.169702

0.010109

0.024982

0.000348

159.0658

2.188191

15

68.12095

661.1316

1358.808

0.048179

0.002254

0.159421

0.007021

0.024178

0.000275

154.0101

1.729854

16

46.29636

489.1592

875.4446

0.053796

0.002973

0.174047

0.00894

0.023738

0.000268

151.2355

1.688236

18

72.21149

668.9532

1304.427

0.05003

0.002163

0.165644

0.007092

0.024052

0.000269

153.2138

1.6907

Pb

Th

U

207Pb/206Pb

207Pb/206Pb

207Pb/235U

207Pb/235U

206Pb/238U

206Pb/238U

206Pb/238U

206Pb/238U

XTL-340

ppm

ppm

ppm

Ratio

1sigma

Ratio

1sigma

Ratio

1sigma

Age (Ma)

1sigma

1

162.7584

1327.97

2786.917

0.061992

0.00208

0.216541

0.007261

0.025216

0.000261

160.5397

1.639684

2

112.4008

804.2761

2622.732

0.052432

0.001812

0.184311

0.00631

0.025386

0.000238

161.6042

1.494933

3

156.5895

1286.701

3143.719

0.050913

0.001425

0.181325

0.005049

0.025704

0.000236

163.6059

1.481032

4

289.9565

4001.691

2070.173

0.063124

0.001981

0.221424

0.007151

0.025267

0.000242

160.8568

1.522363

5

125.0608

983.9794

2797.928

0.04971

0.001495

0.174279

0.005068

0.025412

0.000224

161.7666

1.40774

6

162.3233

1400.659

3221.655

0.050923

0.001605

0.183729

0.00577

0.026005

0.000212

165.4973

1.329999

8

126.8413

1054.429

2488.264

0.05176

0.001811

0.186046

0.00659

0.026183

0.00036

166.6115

2.261453

9

117.7929

909.4558

2589.701

0.04697

0.001462

0.164988

0.005161

0.025367

0.000238

161.487

1.494871

10

185.7832

1665.509

3407.976

0.05224

0.001583

0.183149

0.005569

0.025345

0.000287

161.3463

1.805077

12

183.1203

1590.276

3739.657

0.04901

0.001424

0.171262

0.005023

0.025193

0.000199

160.3941

1.253843

13

172.2477

1509.76

3400.123

0.051271

0.001511

0.179143

0.00522

0.025252

0.000211

160.7609

1.326957

16

401.497

5867.655

3701.837

0.049571

0.001439

0.170975

0.004864

0.024963

0.000221

158.9448

1.389775

18

111.6549

926.5998

2144.36

0.052337

0.00177

0.183989

0.006069

0.025476

0.000214

162.1699

1.345771

Reviewer 3 Report

Report on the manuscript minerals-1731436

"Zircon U-Pb Ages and Geochemistry of the Granite in the Xintianling Tungsten Deposit, SE China: Constraints on Metallogenesis

 by Yang Wu, Zhang Min, Yan Jun and Chen XiaoCui

Dear Authors,

The paper is well-written and the English-language is essential and adequate.

The new results and discussion are correctly integrated with previous information.

I appreciated your research work although there are some fundamental points in which to intervene:

-        The paper is not properly organized;

-        It is necessary to add a figure (number 2?) showing study rocks in outcrop;

-        Probably it was an oversight but it is strictly necessary that you attach the U-Pb data tables as supplementary material available online, also accompanied by the analyses carried out on the standards;

-        It is necessary to implement the figures captions;

-        The number of cited papers is quite small for an international Journal.

Please implement it with international literature references (some example are listed below).

SUGGESTIONS TO IMPROVE THE MANUSCRIPT

I have few suggestions to improve the data presentation.

·       I suggest the following organization of the paragraphs in order to keep the different parts of the manuscript distinct:

1. Introduction

2. Geological framework

2.1 Regional Geology

2.2 Geologic Features of the Deposit (include in this paragraph the brief information about sampling currently found at the beginning of paragraph 4)

Please, add a dedicated Figure (2) showing the studied rocks in the outcrop. This is an essential aspect!!!

3. Experimental Methods

4. Results

4.1 Geochemistry

4.1.2 Major Elements

4.1.3 Trace Elements

4.1.4 REE

4.2 Zircon data

4.2.1 Zircon imaging and chemistry

4.2.2 Zircon U-Pb Geochronology

Please, I advise you to keep separate the data concerning the study of the internal textures, the trace element characterization and finally the isotopic data !!!

5 and 5.1…..Discussion organization is ok

Also for the Conclusions (par. 6)

·       The characterization of internal pattern zoning of zircon crystals deserves more attention and detail (in the paragraph 4.2.1).

Please refer to the paper:

Corfu, F., Hanchar, J. M., Hoskin, P. W., & Kinny, P. (2003). Atlas of zircon textures. Reviews in mineralogy and geochemistry53(1), 469-500.

and

Fornelli, A., Piccarreta, G., Micheletti, F. (2014). In situ U-Pb dating combined with SEM imaging on zircon–an analytical bond for effective geological reconstructions. Geochronology–Methods and Case Studies, 109-39, Edited by Mörner, N. A.

·         For better constrained the significance of Th/U ratios (add in par. 4.2.1), please refer to these papers:

Rubatto, D. Zircon: The Metamorphic Mineral. Rev. Mineral. Geochem. 2017, 83, 261–295.

and

Fornelli, A., Micheletti, F., & Piccarreta, G. (2016). Late-Proterozoic to Paleozoic history of the peri-Gondwana Calabria–Peloritani Terrane inferred from a review of zircon chronology. SpringerPlus-Volume 5, Issue 1, Article number 212, Pages 1-19.

·       The indication of the percentage of U-Pb discordance and the considered cut off used for the age filtering are recommended (in the paragraph 4.2.2).

Also in the U-Pb data table (it is necessary to attach the table as supplementary material!!!) the %U-Pb discordance and U-Pb data for the standards must be indicated following the recent paper published in Minerals (see supplementary material files):

Fornelli, A., Festa, V., Micheletti, F., Spiess, R., & Tursi, F. (2020). Building an Orogen: Review of U-Pb Zircon Ages from the Calabria–Peloritani Terrane to Constrain the Timing of the Southern Variscan Belt. Minerals, 10(11), 944.

·       Figure captions:

-        Figure 2: I invite you to rewrite the caption keeping separate the description of photographs of the macroscopic samples from the textures highlighted under the polarized light microscope

Try to lighten the images and enlarge the red characters (white seems to be preferable) of the mineral abbreviations

-        Figures 3, 4, 5 : I invite you to rewrite the captions following this example:CL images of selected dated zircons; red circles indicate the spot location of concordant ages”

In conclusion,

I think that this paper and results are appealing, and the manuscript can be accepted for publication with major revisions in an international journal as Minerals and only after viewing the U-Pb data Tables.

Author Response

Dear reviewer:

Thank you very much for your guidance and advice. We have learned much from your and other two reviewers’ comments, which are fair, encouraging and constructive. After carefully studying the comments and your advice, we have made corresponding changes. The main revisions are listed below.

(1)According to the comments and contents of article,The title of the article has been amended.We have changed a scientific and reasonable titles: Zircon U-Pb Ages and Geochemistry of the Granite in the Xintianling Tungsten Deposit, SE China: Implications on geodynamic settings of the regional tungsten minerlization.

(2)The Abstract section was rewritten to highlight the conclusions as suggested by the reviewers.

(3)According to the reviewer's request, Regional Geological Characteristics was rewritten,Corresponding to the topic, the regional structure is mainly introduced

(4)The Discussion section has been rearranged in response to scientific issues raised in Introduction. 

(5)Supplement the lack of standardization and standard sample part of the test

(6)References were renumbered and optimized.Due to forgetting to adjust the reference when revising the article(Deleted a lot of content.), there was confusion in the reference labeling, which has been corrected.

(7)We supplement the original U-Pb zircon data and discard the Concordia map of U-Pb which have problems 

(8)We have polished the language with the help of MDPI official.

(9)Some figures have been rearranged to avoid contradiction. Some approaches have been reconsidered to make estimation and calculation more convinced and accurate.

(10)Some contents have been cut to make the manuscript more concise, as suggested by the reviewers.

More detailed revisions are annotated in the revised manuscript. (As required, I used the modification mode in Word)

Round 2

Reviewer 1 Report

I found the text improved in the various points highlighted in my previous review; new figures and tables have been added, and the English language is now fine too. In my opinion the paper is now ready for publication.

Author Response

Dear reviewer:

Thanks again for your guidance and advice. We have learned much from your and other two reviewers’ comments.

best wishes to you

Kind regards,

Yang Wu

Reviewer 2 Report

minor checks directly on the manuscript.

the only major point is to add the wetherill and relative mean age plots for each of the three age groups.

Author Response

Dear reviewer:

Thanks again for your guidance and advice. We have learned much from your and other two reviewers’ comments, we have made corresponding changes. The main revisions are listed below.

Response to Reviewer 2

  1. what do you mean here? How can the crust have melted the granitoids? you talking of the influence of crustal thickening?

Revision:Yes, the paragraph was ambiguous, so I shortened the sentence and rewrote it

The first school of thoughts proposes that granite in the belt is closely related to S-type granite , whereas the others thought that the exposed granite is primarily A-type granite.

  1. rephrase:Our new results shown here, when combined with other independent evidence, demonstrate that the Xintianling tungsten deposit and regional mineralization formed in late Jurassic under a back-arc extensional setting

Revision:Our new results shows that the Xintianling tungsten deposit  formed in late Jurassic under a back-arc extensional setting. The events were likely triggered by the break-off or detachment of the flatly subducted Paleo-Pacific slab.

  1. what do you mean bytectonomag-matic zone? how is it any different from a magmatic zone? Explain

Revision:Because the area has ex-perienced many complicated tectonic movements in different stages and  accompanied by magmatic activities.The structure restricts the magmatism,So I choose this word: tectonomag-matic zone.

  1. either useless definition, or you may mean that you have evidences of the chronology of the shear zone taking place before the magmatic activity and that the second is deeply related to the accomodation space created in the crust by the strike shear.if the last is the case than you have to support this better

 Revision: Delete:which is made up of a fracture zone and intruded magma

  1. Other issues noted in the PDF have been revised accordingly

6.the only major point is to add the wetherill and relative mean age plots for each of the three age groups.

Revision:  increased  

Reviewer 3 Report

Dear Authors,

I believe that the revision of the original manuscript was rather hasty omitting numerous suggestions that were recommended after the first reading.

I have again indicated the necessary changes in my opinion and I hope this time they will be implemented.

Best regards

Author Response

Dear reviewer:

Thanks again for your guidance and advice. We have learned much from your and other two reviewers’ comments, we have made corresponding changes. The main revisions are listed below.

Response to Reviewer 3

 The organization of the paragraphs was only superficially changed

Revision:Some executive judgments are deleted in the result section,Other chapters have been cut to highlight themselves

2-Figures 2 and 3 have been usefully added on the geological section, but exhaustive field photos showing studied rocks in outcrop are still missing

Figure 4 shows only the macroscopic samples from which the thin sections of rock were probably obtained!!! Kindly brought the necessary corrections.

Revision:Added  granite field photos

3The data collected on zircon crystals regarding internal pattern zoning, chemistry and age results are presented in a single paragraph together with the analytical conditions!!!The indication of the percentage of U-Pb discordance and the considered cut off used for the age filtering are not indicated! In the U-Pb data table the %U-Pb discordance and U-Pb data for the standards must be added!!!Please, add all these data.Because the Concordia diagrams have been deleted?

Revision:In the first review, one reviewer pointed out that there were some fluctuations in U-Pb data shown in Concordia diagrams.he recommended to supplement the original data and add standard sample instruction replace Concordia diagrams.This time I added again, and add the std sample data in the table

4.Please, verify the number of figure; actually they are 12 and not 11.

Revision:It has been modified

5-Added references 29-30-31 and 32 are improperly cited.

Please, insert they in the appropriate positions (Rubatto and Fornelli 2016 for U-Th ratio interpretation; Corfu for internal zoning description; Fornelli 2020 for U-Pb data processing)

Revision:It has been modified